# 5'UTR G-quadruplex structure enhances translation in size dependent manner

Chun-Ying Lee[1,3], Meera Joshi[1,3], Ashley Wang [1] & Sua Myong [1,2] ✉

Translation initiation in bacteria is frequently regulated by various structures in the 5' untranslated region (5'UTR). Previously, we demonstrated that G-quadruplex (G4) formation in non-template DNA enhances transcription. In this study, we aim to explore how G4 formation in mRNA (RG4) at 5'UTR impacts translation using a T7-based in vitro translation system and in *E. coli*. We show that RG4 strongly promotes translation efficiency in a size-dependent manner. Additionally, inserting a hairpin upstream of the RG4 further enhances translation efficiency, reaching up to a 12-fold increase. We find that the RG4-dependent effect is not due to increased ribosome affinity, ribosome binding site accessibility, or mRNA stability. We propose a physical barrier model in which bulky structures in 5'UTR biases ribosome movement toward the downstream start codon, thereby increasing the translation output. This study provides biophysical insights into the regulatory role of 5'UTR structures in in vitro and bacterial translation, highlighting their potential applications in tuning gene expression.

Gene expression is a tightly regulated process to ensure efficient utilization of resources and adaptation to changing environments. This regulation occurs at various levels, including transcription, translation, and the level of mRNA and protein[1–7]. In bacteria, the absence of a nuclear membrane necessitates rapid post-transcriptional regulations to enable quick responses to environmental stimuli[8–11]. Untranslated regions (UTR) of RNA have emerged as key players in regulating translation initiation by presenting noncanonical structures to translational machinery or by recruiting proteins and enzymes that recognize RNA sequences, modifications, or structures[12,13].

The 5' untranslated region (5'UTR) of bacterial mRNA serves multiple functions that are critical for gene regulation and protein synthesis. First, it typically contains a conserved AG-rich Shine-Dalgarno (SD) sequence, located a few nucleotides upstream of the translation start site (TSS). The SD sequence base-pairs with the 16S ribosomal RNA (rRNA) to guide the binding of the small ribosomal subunit, providing a well-defined mechanism for initiating translation[14]. Second, bacterial 5'UTRs often harbor cis-acting regulatory elements, such as upstream open reading frames (uORFs), which stall the ribosome and control the access to downstream TSS. Additionally, bacterial 5'UTRs can serve as a platform for RNA-binding

proteins and small RNAs that regulate translation efficiency. For example, small RNA coupled with an RNA binding protein Hfq can bind a 5'UTR to stimulate translation initiation or trigger mRNA degradation[10,15,16].

Furthermore, secondary structures within 5'UTR play a critical role in regulating RNA stability and translation efficiency. Specifically, co-transcriptionally folded structures can influence translation initiation and the rate of translation. Previous research on bacterial 5'UTR primarily focused on the ribosome binding site (RBS), which includes the SD sequence and a short range (10–20 nt) upstream and downstream of the SD region[14,17]. Several studies showed that secondary structures, such as pseudoknots and hairpin stem-loop that form across the SD sequence can inhibit translation by preventing ribosome binding[18,19]. Notably, temperature-sensitive hairpins in the 5'UTR of *E. coli* can regulate translation by masking or unmasking RBS or start codon (AUG) to turn on or off translation initiation, respectively[20]. Similarly, riboswitches control translation through changes in mRNA conformation upon ligand binding, enabling rapid responses to environmental cues[21,22]. Specific sequence elements, such as purine-rich regions or G-quadruplexes, can also affect translation in a context-dependent manner. Previous studies demonstrated that placing an

[1]Department of Biophysics, Johns Hopkins University, Baltimore, MD 21218, USA. [2]Physics Frontier Center (Center for Physics of Living Cells), University of Illinois, Urbana, IL 61801, USA. [3]These authors contributed equally: Chun-Ying Lee, Meera Joshi. ✉e-mail: sua.myong@childrens.harvard.edu

RNA G-quadruplex (RG4) structure located near the SD sequence inhibits translation by interfering with the base pairing between 16S rRNA and the mRNA[23,24]. Although the studies revealed the effect of RG4 depends on the inserted location and orientation, it remains unclear how and to what extent the RG4 on 5'UTR impacts bacterial gene regulation.

Our previous research has uncovered the role of potential G-quadruplex sequence (PQS) in non-template DNA in promoting transcription through co-transcriptional formation of R-loop and G4 structure[25]. In this context, the transcribed mRNA bears a G4 structure at the 5' end, which prompted us to investigate whether such G4 structures in RNA modulates translation outcome. G-quadruplexes form in single-stranded DNA or RNA that harbors repetitive runs of guanines interspersed with non-guanine, loop sequences. Four guanine bases come together in a coplanar arrangement to form a tetrad, which stacks in multiple layers. The size and stability of the structure depend on the composition and the length of the loops[26]. Increasing evidence suggests that RG4 structures are involved in translation regulation in eukaryotes, often blocking translation initiation when present in the 5'UTR[27–31]. Some RG4 structures have also been shown to function as Internal Ribosome Entry Sites (IRES), stimulating translation independent of a start site[32–34]. In addition, despite the low abundance of PQS in prokaryotes, computational studies have identified a few conserved G4 motifs across prokaryotic species positioned non-randomly in promoter and intergenic region, indicating an evolutionarily conserved function of G4 in bacterial genome[35–37]. A recent transcriptome analysis finds that most folded RG4 in bacteria are two-quartet G quadruplex and mainly detected in the coding region and underlines the function of RG4 up- or down-regulating gene expression differs between species[38].

In this study, we investigated the potential role of 5'UTR RG4 structures in *E. coli* translation. We inserted a series of PQSs in non-template DNA, upstream of a GFP reporter gene to allow for the formation of RG4 at 5'UTR. Using the T7 expression system, we measured in vitro transcription and subsequent translation in real-time, which were used to calculate the translation efficiency. We found that the presence of RG4 in the 5'UTR led to enhanced translation both in vitro and in *E. coli*. Longer loops within RG4 resulted in higher translation yield. Moreover, insertion of a hairpin upstream of an RG4 further increased translation. Taken together, we demonstrate that the translation enhancement scales with the size of the 5'UTR structures. We propose a mechanism by which the 5'UTR structures act as a physical barrier that may bias ribosome movement toward the downstream region and thereby promote translation.

## Result

### RNA G4 increases translation efficiency

To quantify the translation efficiency, we set up an in vitro translation assay which contains reagents for the T7 RNA polymerase (RNAP) and *E. coli* translation system (Fig. 1a)[25,39]. We prepared DNA construct with T7 promoter followed by the ribosome binding site (RBS), a fluorescent reporter which encodes *superfold* GFP (*sf*GFP), and a transcription terminator sequence. We used our previously established protocol to measure the real-time transcription; DNA molecular beacon becomes fluorescent when annealed to a transcribed RNA which bears a complementary sequence (Fig. 1b)[40]. In parallel, the intensity of *sf*GFP was obtained and plotted as a translation readout (Fig. 1c). Hence, the real-time transcription and translation activities can be simultaneously measured by collecting intensities of the molecular beacon and GFP over time using a plate reader. The GFP signal is expected to rise after that of the molecular beacon because of the time delay between transcription and translation and the maturation time required for the *sf*GFP folding[41]. Based on the simultaneous measurement, we can calculate the translational efficiency for each reaction by normalizing the translation signal (*sf*GFP) by the transcription signal (molecular beacon).

To examine the effect of 5'UTR RNA G-quadruplex (RG4) on translation, we inserted a potential G-quadruplex forming sequence (PQS) in between the T7 promoter and the RBS such that the PQS is 44 bp downstream of T7 promoter and 43 bp upstream from the RBS. The PQS was inserted into either a template (T) or a non-template (NT) strand for comparison. Hence, the PQS insertion in NT is expected to produce the G4-bearing transcript which can fold into RG4 while the PQS in T, and the scrambled control (C) sequence will not fold into RG4 (Fig. 2a). We also note that the construct was derived from an expression plasmid, which contains a 10 bp stem loop at 5' end as the sequence of *lac* operon. In agreement with our previous study, the PQS-NT led to approximately 30% higher transcription efficiency than in PQS-T (Fig. 2b)[25]. The initiation rate of transcription was quantified by taking the linear increase of fluorescence intensity in the initial phase of each curve (Fig. 2c). Surprisingly, the translation reporter, *sf*GFP signal revealed that the PQS-NT induced over fivefold higher protein product compared to the control (Fig. 2d), which is not related to the presence of molecular beacon (Supplementary Fig. 1a). The translation efficiency (TE) was calculated by dividing the initiation rate of *sf*GFP signal by the transcription initiation rate for each condition. (Fig. 2e). The 5-fold difference observed in translation cannot be explained by the 30% difference in transcription between the NT and T, suggesting an additional mechanism that promotes translation post-transcriptionally. Due to the PQS orientations, we expect that the

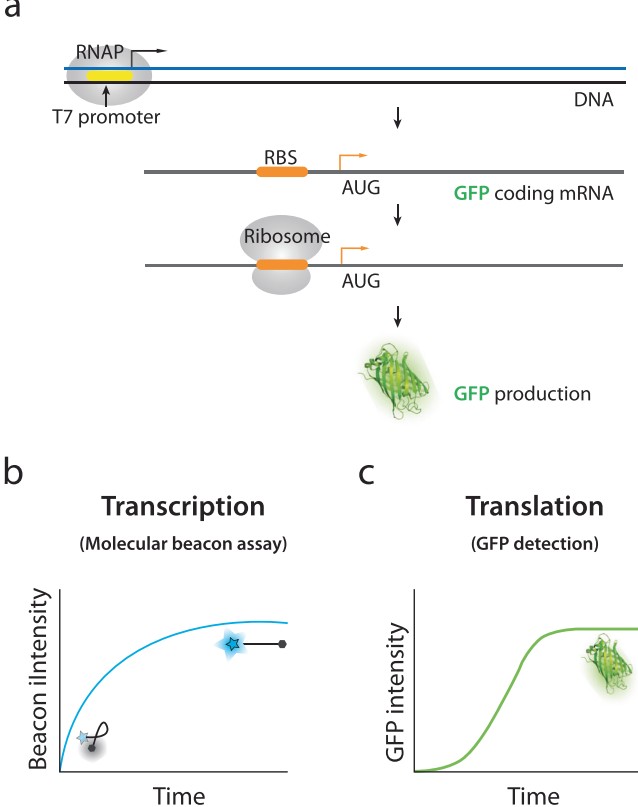

**Fig. 1 | Scheme of in vitro co-transcriptional translation assay and quantification. a** Schematic of the co-transcriptional translation assay. The DNA construct features a T7 promoter (yellow), *E. coli* ribosome binding site (RBS in orange) and GFP gene. **b**, **c** Example curves of transcription and translation data. The blue curve exemplifies the transcription readout in real-time via a Cy3 molecular beacon that fluoresces when bound to mRNA. The green curve represents translation readout as the fluorescence of *sf*GFP. The initial linear phases of curves are used to quantify the transcription and translation initiation rate, respectively. The calculation is described in "Method".

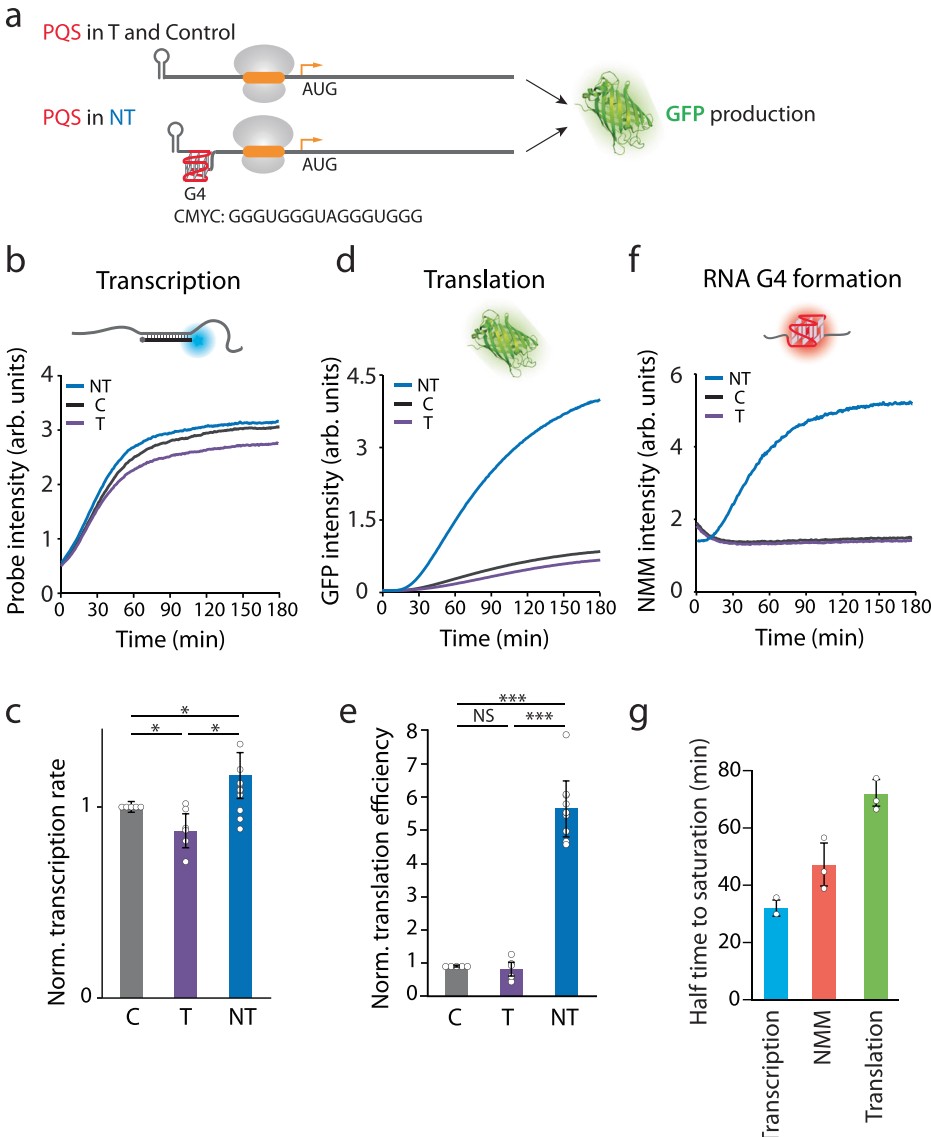

**Fig. 2 | PQS orientation and presence of RNA G4 increases translation efficiency. a** Schematic of RNA G4 insertion in the 5′UTR. The potential G-quadruplex sequence (PQS) in non-template (NT) strand leads to the formation of RG4 in the mRNA, while insertion in template (T) strand and control (C) does not. **b, d, f** Real-time intensities of transcription, translation and G4 formation assays. The constructs, non-template (NT), control (C), and template (T) are colored in blue, black, and purple, respectively. The curves shown are one representative result from multiple independent experiments. The control and tested PQS sequences are scramble sequence and cMyc. **b** Transcription assay is quantified by Cy3 probe intensity. **d** Translation is measured as *sf*GFP intensity. **f** RNA G4 formation is quantified by real-time N-methyl mesoporphyrin IX (NMM) signal. NT shows NMM signal while C and T have no signal. **c** Transcription rates are calculated from the early linear part of the curve in (**b**) and normalized to the transcription rate of the control sequence. The transcription rate of NT is 30% higher than T. **e** NT construct enhanced the translation fivefold higher than C and T. Translation efficiencies are calculated from the translation rates obtained from the early linear part of the curve in (**d**) and the normalized transcription rates in (**c**). The translation efficiency was normalized to the control sequence. For (**c**) and (**e**) data are presented as mean ± SEM of independent experiments ($n > 6$). Exact mean values are provided in Supplementary Tables 3.1 and 3.2. NS: nonsignificant, *$P < 0.05$, ***$P < 0.0005$ (two-sided unpaired *t* test). **g** Halftime to saturation of NT. The halftimes of transcription (in **b**), translation (in **d**), and NMM RG4 formation (in **f**) are 32 ± 2.8, 47 ± 7.5, and 71.7 ± 4.6 min, respectively. Data are presented as mean ± SEM of independent experiments ($n = 3$). Raw data points are provided as a Source Data file.

mRNA from NT, but not T or control forms an RG4 structure at 5′UTR. Thus, we tested for the RG4 formation by applying N-methyl meso-porphyrin IX (NMM) in the transcription reaction[42]. NMM is a G4 ligand that exhibits induced fluorescence upon binding G4[43]. As expected, the NMM fluorescence displayed a prominent increase over the transcription time in NT, but not in T and C (Fig. 2f), indicating a progressive and robust formation of RG4 exclusively in NT condition. The selective NMM fluorescence for NT suggests that RG4 is likely responsible for the enhanced translation since the three constructs differ only by the PQS region. In addition, the half-life obtained from

the fluorescence increase reflected the order of events i.e., transcription signal increased first, followed by the NMM intensity reflecting the RG4 formation, and the GFP intensity (Fig. 2g). This further supports that the RG4 is responsible for the enhanced translation.

## Bulkiness of RG4 drives translational enhancement

PQS can vary in its sequence composition, which gives rise to diverse conformations of varying stability and bulkiness[42,44]. Based on the result obtained above, we asked if different PQS sequences produce various levels of translational enhancement. To focus on the effect of

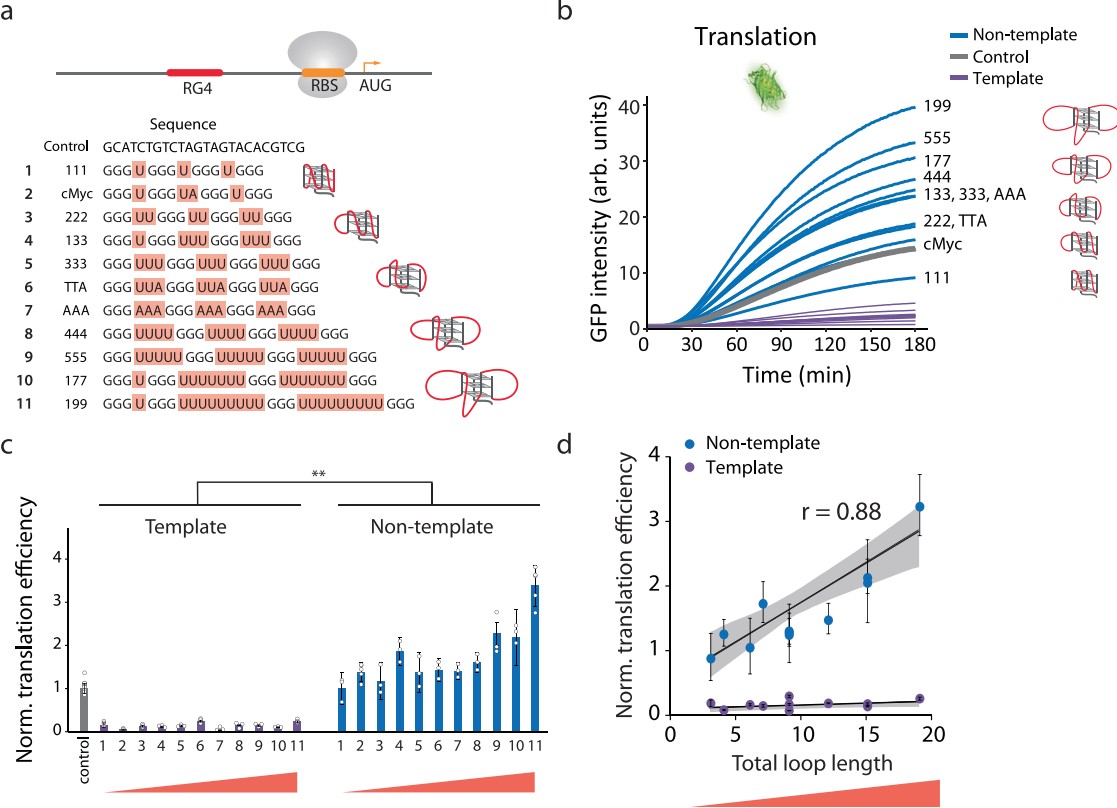

**Fig. 3 | Bulkiness of RG4 drives translational enhancement. a** Potential G-quadruplex sequence (PQS) candidates with varying loop lengths, representing RNA G-quadruplex (RG4) in RNA. The core domain of guanine triplets (black) remains constant, and loop sequences (highlighted in red) are varied. The sequences are arranged from shortest to longest loop length, indicating an increase of bulkiness. **b** Real-time GFP signal measurements of individual PQS candidates by plate reader. Non-template (NT), Template (T), and control (C) are colored in blue, purple, and gray, respectively. The curves shown are one representative result from multiple independent experiments. **c** Normalized translation efficiencies are cal-culated from the early linear part of curve in (**b**). The numbers on the x-axis represent the PQS ordered as shown in (**a**). NT constructs with longer loop lengths, representing bulkier RG4 structures, resulted in higher translation efficiencies. **d** The correlation between translation efficiency and total loop length. The loop length is the sum of the uridine bases in (**a**). The correlation coefficients are 0.88 and 0.36 for NT and T, respectively, indicating a strong correlation between translation efficiency and RG4 for NT. Data are presented as mean ± SEM of $n = 3-5$ and three independent experiments for NT and T, respectively. Exact mean values are provided in Supplementary Table 3.3. Raw data points are provided as a Source Data file. Shown in (**c**) only represents the significance between template and non-template, where **$P < 0.005$ (two-sided unpaired $t$ test).

G4, we removed the hairpin structure upstream of PQS and inserted a series of PQS with varying loop lengths without changing the guanine triplets (Fig. 3a). Since RNA G4 primarily folds in parallel conformation in which all the guanine strands run in the same orientation[45], we envision that the loop sequences will protrude out from the central tetrad core. The NMM-based assay revealed that transcription of all PQS-NT sequences produced RG4 (Supplementary Fig. 1b–d). Despite varying levels of NMM signal acquired for different RG4s, single-molecule FRET assay displayed that both small (short looped) and large (long looped) RG4 form a stable G4 structure without structural dynamics (Supplementary Fig. 1e–g). Furthermore, RG4 formation of small (cMyc) and large sized (199) PQS was confirmed by NMR spectra (Supplementary Fig 2a), and the thermal stabilities of the RG4s were probed by the reverse transcriptase stop assay, generating truncated cDNA products (Supplementary Fig 2b). Surprisingly, without the upstream hairpin structure all PQS-NT which result in RG4 containing RNA consistently led to higher translation level than control and its counterpart PQS-T construct (Fig. 3b and c), strongly reflecting the role of RG4 in promoting translation. Next, to examine the relationship between the RG4 sequence and the translation level, we plotted the translation efficiency against the total loop length of each RG4. Strik-ingly, the loop length of RG4 is highly correlated to the translation efficiency with a correlation coefficient of $r = 0.88$, indicating that the longer loop length which likely represents higher bulkiness of

individual RG4 structure drives translation enhancement (Fig. 3d). Overall, our data demonstrate that all RG4 structures elevate transla-tion and the bulkiness of RG4 accentuates the translational enhancement.

## Hairpin and RG4 structures synergistically promote translation

By comparing the results presented in Figs. 3c and 2, we noticed that despite the same PQS (cMyc) sequence, the translation enhancement was higher in Fig. 2. Upon close examination, we hypothesized that the 10 bp hairpin forming sequence located upstream of PQS in the con-struct used in Fig. 2 may contribute to the enhancement. This obser-vation led us to test if an additional 5′UTR structure can further enhance translation. To investigate the effect of two tandem struc-tures, we divided the 5′UTR into four segments (Fig. 4a and Supple-mentary Table 1): upstream of RG4 (1), RG4 (2), downstream of RG4 (3), and RBS to start codon (4). We applied an RNA structure prediction tool (*UNAfolds*[46]) to calculate the folding energies of all positions except position 2, because RG4 folding cannot be accurately predicted by currently available tools. The folding energy, $\Delta G$ estimated for the positions 3, 4, and 3 + 4 were −5.6, 0.9, and −11.8 kcal/mol, respectively, indicating weakly folded state of the downstream sequence (Supple-mentary Table 1). To avoid interference with the ribosome binding, we decided to vary sequence located upstream of PQS. We note that the position 1 sequence used in Fig. 3 has a low folding energy ($\Delta G = -3.1$

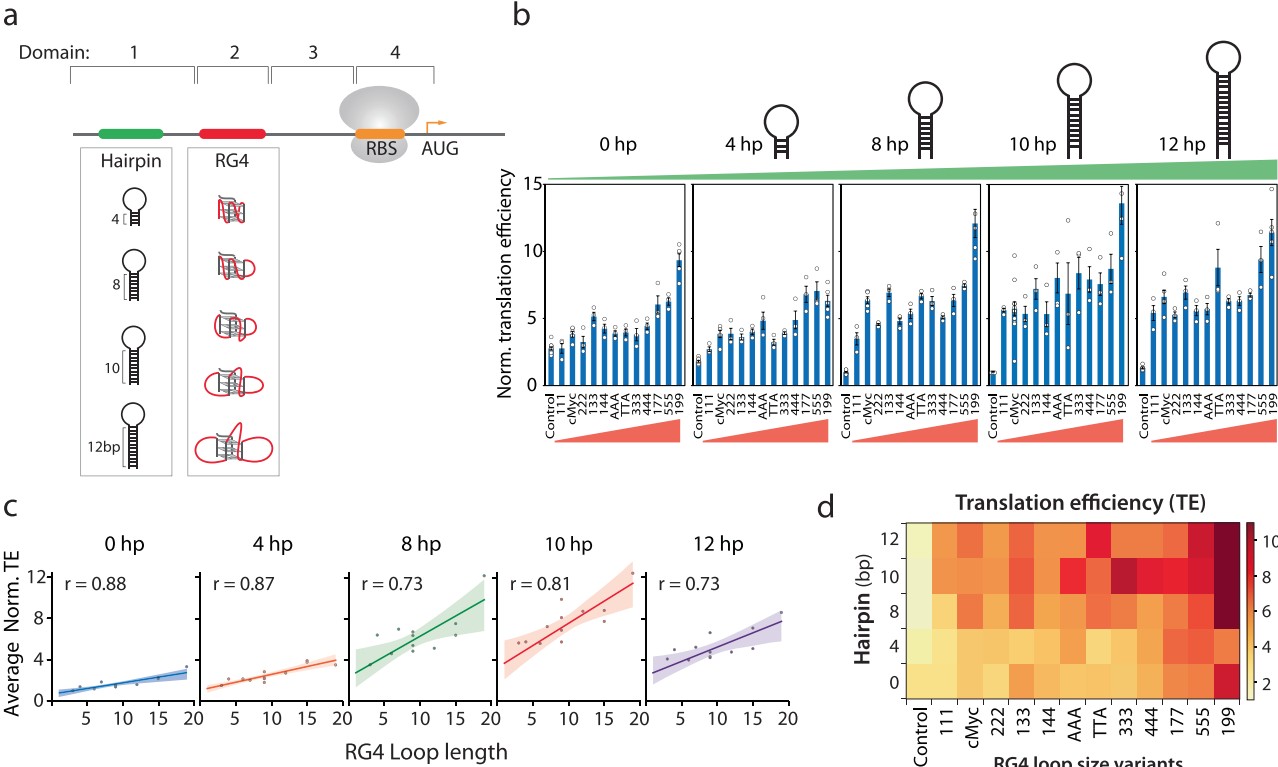

**Fig. 4 | Hairpin and RG4 structures synergistically promote translation.**
**a** Schematic of the 5′UTR. Domain 1 is upstream of the 5′UTR and the position for hairpin insertion. Domain 2 is the potential G-quadruplex sequence (PQS) location. Domain 3 is downstream of PQS. Domain 4 begins at the ribosome binding site (RBS) and ends at +9 of translation codons. The predicted mRNA folding energies are provided in Supplementary Table 1. **b** Translation efficiencies of non-template constructs containing different hairpins (hp) and RNA G-quadruplex (RG4) in the 5′ UTR. The constructs are grouped according to hairpin stem length, ranging from short (left) to long (right). The 0 hp data set was copied from Fig. 3c and normalized to 10 hp-control. The x-axis of each group indicates the RG4 sequence in domain 2, arranged in increasing total loop length. Translation efficiencies for all constructs are normalized to the 10 hp-control. Data are represented as mean ± SEM of

$n = 3$–16. Raw data points and exact mean value are provided as a Source Data file. **c** The correlation between translation efficiency and total loop length. The data are grouped according to hairpin stem length, and each data point is normalized to the control within the group. Strong correlations are observed between translation efficiency and loop length for all hairpin structures, indicating that dependence on domain 2 RG4 remains regardless of domain 1. The regression line was plotted with 95% line confidence band. **d** Heat map of the mean translation efficiencies from (**b**). The x-axis is RG4 arranged in order of longer loop lengths, and the y-axis is the length of hairpin stem. The color represents the mean translation efficiency and is scaled from 0 (white) to 12 (dark red). The color trend demonstrates the synergistic dependence of hairpin and RG4 size on translation efficiencies.

kcal/mol) (see Supplementary Table 1), which most likely stays unfolded at 37 °C; we named the construct 0 hp henceforth. To study the impact of hairpin in modulating translation, we introduced hairpins of 4 bp, 8 bp, 10 bp and 12 bp stem length which are expected to have folding energy ($\Delta G$) of −5.9, −10.4, −15.9, and −21.4 kcal/mol, respectively (Fig. 4a). We inserted each hairpin at position 1, upstream of the RG4 sequences. Following the trend seen in 0 hp cases (Fig. 3), the translational efficiency for each hairpin group exhibited an RG4 size dependence (Fig. 4b, c), indicating that the RG4 structures remained in different hairpin constructs. In addition, the translation was further enhanced as a function of the hairpin length (Fig. 4b). For example, the translation of cMyc increased from 1.4 to 5 folds for 0 hp and 12 hp, respectively and that of 199 increased from 3.3–8.5 folds for 0 hp and 12 hp, respectively. However, none of the hairpins enhanced translation by itself in the absence of RG4 (the control of each hairpin), suggesting that single hairpin structure cannot drive the translation enhancement independently. Next, we tested tandem hairpins by placing 6, 10, 14, and 17 hp (folding energies $\Delta G$: −6.5, −21.0, −31.3, −42.1 kcal/mol, respectively) in addition to the 10 hp (Supplementary Fig. 3). We found that the two hairpins enhance translation only to a level of a single RG4 (cMyc) regardless of the folding energy (Supplementary Table 1). This suggests that the enhancement was induced by the type of structure rather than the folding energy and that the RG4 is more potent than the hairpins in promoting translation. Indeed, the

RG4 loop length-dependent translational efficiency is exhibited for all hairpin variants (Fig. 4c). We compiled and projected all the results to a 2-D heatmap which presents a distinct trend that translation strength is highly correlated with both the hairpin stem size (vertical axis) and G4 loop lengths (horizontal axis) (Fig. 4d). Taken together, hairpin contributes to enhanced translation only when present with RG4. This finding raises an intriguing question about the underlying mechanism by which 5′UTR structure upstream of RBS enhances translation.

What is the mechanism that enables 5′UTR structures to elevate translation in a size-dependent manner? We reasoned that the 5′UTR structure may have an impact on either the ribosome or the mRNA. To define the mechanism, we set out to test four hypotheses: i) RG4 increases the ribosome binding affinity (Fig. 5a); ii) RG4 improves the accessibility of RBS to ribosome (Fig. 5e); iii) RG4 increases the mRNA lifetime (Fig. 6a); iv) RG4 stabilizes the ribosome-bound state (Fig. 6c).

## RG4 does not attract ribosome
We tested the hypothesis that RG4 enhances translation by increasing the affinity to ribosomes, perhaps by acting like an IRES (internal ribosome entry site) in viral translation[47]. This hypothesis posits that the mRNA containing RG4 structure (based on PQS-NT) will recruit more ribosomes than the mRNA without RG4 (Control, PQS-T) (Fig. 5a), resulting in higher translation. To test this hypothesis, we performed a competition assay in which RG4 bearing competitor RNA

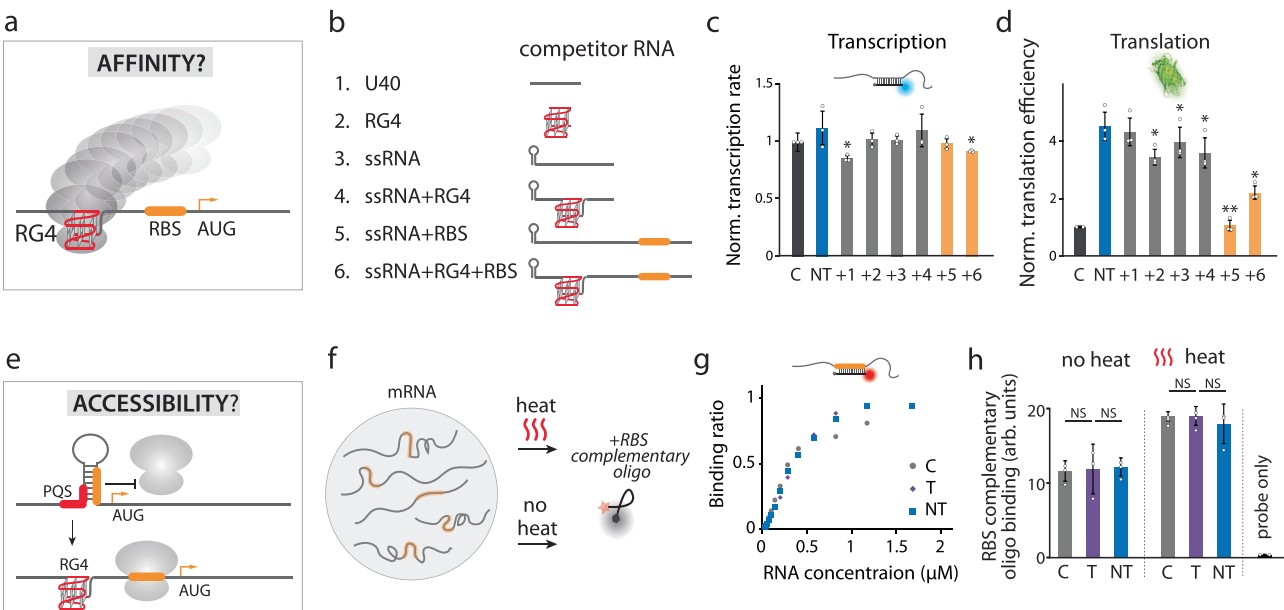

**Fig. 5 | RG4 does not recruit ribosome or increase RBS accessibility.**
**a**–**d** Hypothesis one: RNA G-quadruplex (RG4) increases affinity to ribosome.
**a** Schematic of RG4 recruiting the ribosome. **b** Competitor RNA used in translation reaction. The RG4 is cMyc (Fig. 3a), and the hairpin is 10 hp (Fig. 4a). **c** Normalized transcription rate of cMyc Non-template (cMyc-NT) with addition of competitor RNA. **d** Normalized translation efficiency. Additions of RNA 1–4 show weakly significant difference from cMyc-NT while RBS containing RNA (5 and 6) reduces GFP production, suggesting successful competition requires RBS not RG4. For (**c**) and (**d**), data are presented as mean ± SEM of independent experiments ($n = 3$). Exact mean values are provided in Supplementary Tables 3.4 and 3.5. Raw data points are provided as a Source Data file. Shown in (**c**, **d**) represents the significance between cMyc-NT and addition of competitor, where *$P < 0.05$, **$P < 0.005$ (two-sided paired t-test). **e**–**g** Hypothesis two: RG4 enhances the accessibility of RBS to ribosome.
**e** Cartoon depicting the release of RBS by RG4. **f** Schematic for examining the

accessibility of RBS. The mRNA is either heated to remove all secondary structure or not heated. A Cy5-labled molecular beacon complementary to the RBS is applied to determine accessibility. **g** Binding curves of the RBS molecular beacon to RNAs. The dissociation constants, $K_d$, are 367 (±25), 364 (±23), 370 (±21) nM for Non-Template (blue rectangle), Control (gray circle), and Template (purple diamond), respectively, showing a negligible difference in binding affinity (two-sided paired t-test). The dot curves shown are one representative result from $n = 5$ independent experiments. Raw data points are provided as a Source Data file. **h** Fluorescence intensities of molecular beacon with and without heating. There is no significant difference among the control, T, and NT with or without heating. Data are presented as mean ± SEM of independent experiments ($n = 3$). Exact mean values are provided in Supplementary Table 3.6. Raw data points are provided as a Source Data file. NS nonsignificant (two-sided paired t test).

was applied to the translation reaction in molar excess. The competitor RNA constructs included a negative control, polyU 40 nt (Fig. 5b-1), an RG4 alone (Fig. 5b-2), a single strand (ss) RNA with a hairpin but without RG4 (Fig. 5b-3), an RG4 flanked by the neighboring sequence found in mRNA (Fig. 5b-4), an ssRNA with RBS (Fig. 5b-5) and an ssRNA with both RG4 and RBS (Fig. 5b-6). If our hypothesis is correct, we expect to see reduced translation in conditions 2, 4, and 6, all of which contain RG4. We confirmed that there was no significant difference in transcription rate among the six conditions, indicating that the competitor RNAs did not affect the overall transcription (Fig. 5c). The translation result revealed that the RBS containing RNA (5 and 6) significantly decreased the translation, while the other conditions (1-4) had only a slight effect (Fig. 5d) This suggests that the RBS containing competitor, but not the RG4 bearing strands competed for the ribosome binding, thus lowering the translation. Therefore, we show that the RG4 does not increase affinity toward ribosome.

## RG4 does not increase RBS accessibility
Next, we hypothesized that the formation of RG4 increases RBS accessibility to ribosome by preventing RBS from folding into an inaccessible secondary structure (Fig. 5e). To examine the accessibility of the RBS, we applied a molar excess of a molecular beacon that bears sequence complementary to RBS (Fig. 5f). The beacon fluoresces upon hybridizing to the RBS; hence the intensity of the beacon represents the accessibility of the RBS. We tested the accessibility in two ways. First, we performed titration of purified transcript of NT, C and T to a fixed concentration of molecular beacon (400 nM). The dissociation

constant, $K_d$ was 367 (±25), 364 (±23), 370 (±21) nM for NT, C, and T respectively, indicating negligible difference in RBS accessibility among the three constructs (Fig. 5g). Second, we applied the molecular beacon before and after heating up the transcript from NT, C, and T to test if the heat-induced unfolding will increase the RBS accessibility. Despite the overall increase, the similar intensities of molecular beacon among NT, C, and T indicated that RBS is accessible regardless of RG4 (Fig. 5h). Hence, both assays corroborate to reflect that RBS is fully accessible in all three constructs. Therefore, the RG4 unlikely acts via making the RBS accessible to ribosome loading.

## RG4 does not increase mRNA lifetime
Next, we hypothesized that RG4 increases the lifetime of the mRNA by stabilizing the mRNA (Fig. 6a) since secondary structures on RNA can increase RNA lifetime by preventing RNA degradation[13]. We performed RT-PCR to compare the mRNA from NT, C and T after 3 h of translation. The highly similar mRNA levels tested by two different sets of primers in NT, C and T provide robust evidence that RG4 does not play a role in stabilizing mRNA in our experimental condition (Fig. 6b).

## RG4 may be a physical blockade that biases the ribosome movement toward AUG
The negative results obtained in the first three hypotheses strongly suggest that the effect of RG4 in translational regulation must occur after the ribosome loads on the mRNA. This raises a possibility that RG4 may promote translation by biasing the ribosome movement toward the AUG position (Fig. 6c). To test the hypothesis, we applied

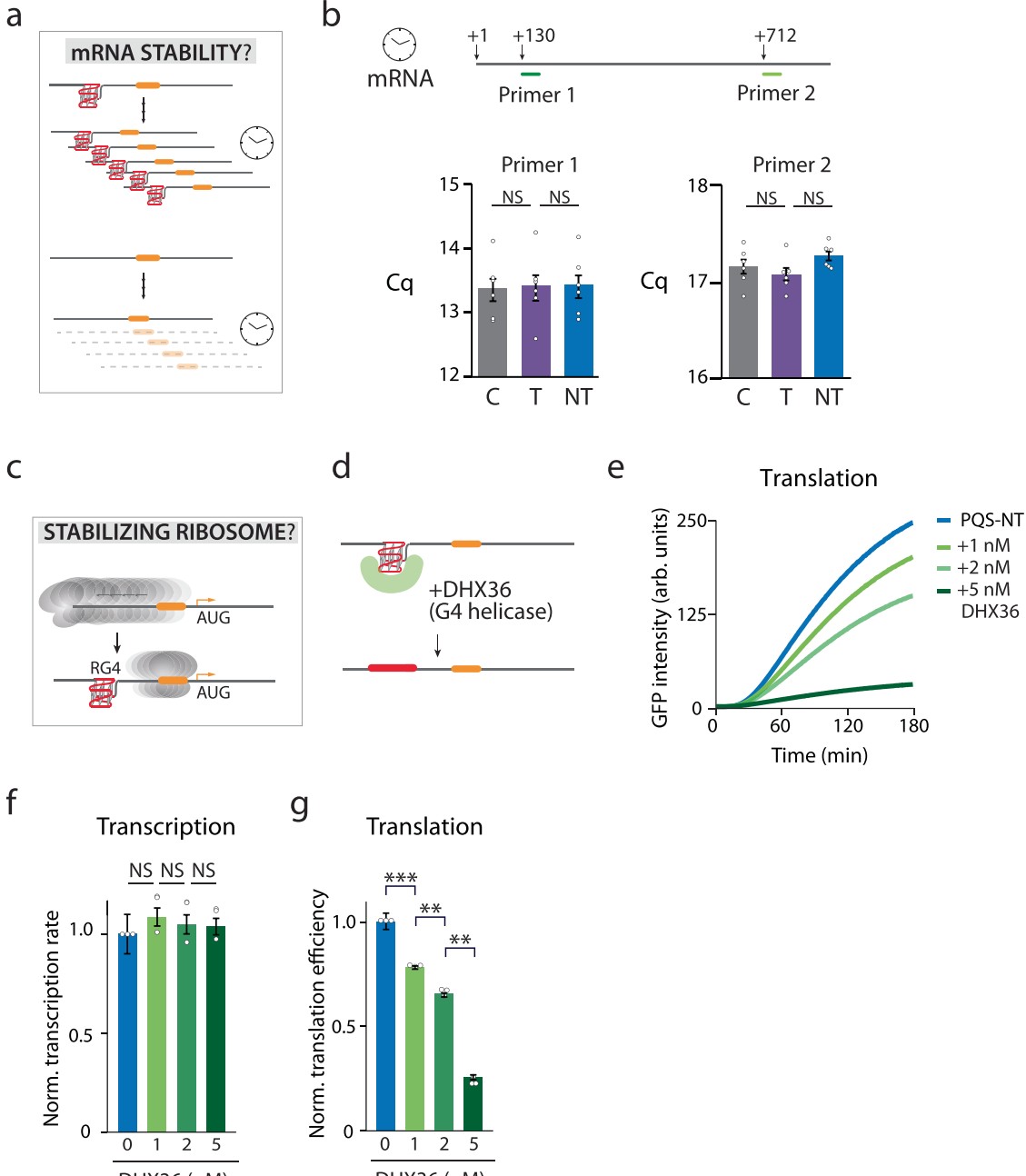

**Fig. 6 | RG4 does not increase mRNA lifetime but may stabilize ribosome bound to mRNA. a**, **b** Hypothesis 3: RG4 increases the mRNA lifetime. **a** Illustration of how RG4 could prevent the mRNA degradation. **b** RT-PCR is performed with two primers, proximal and distal to the 5′ end. No significant difference is observed among the control, Template (T), and Non-template (NT) transcript levels with either primer, indicating that RG4 does not affect mRNA lifetime. Data are presented as mean ± SEM of independent experiments (*n* = 6). NS: nonsignificant (two-sided paired *t* test). Exact mean values are provided in Supplementary Table 3.7. Raw data points are provided as a Source Data file. **c**–**g** Hypothesis 4: RG4 stabilizes the ribosome-bound state. **c** A potential mechanism of RG4 stabilizing the ribosome bound to the mRNA by preventing ribosomes from dislodging off the mRNA. **d** RG4 specific helicase, DHX36, is used to remove RG4 from mRNA during

translation. **e** Translation readout as real-time GFP intensity. The titration of helicase demonstrates a dose-dependence such that the more addition of helicase decreases the translation level further. The curves shown are one representative result from three independent experiments. **f** Normalized transcription rate of NT construct with DHX36 titration. **g** Normalized translation efficiencies are calculated from the early linear part of the curve in (**e**), and normalized to transcription level in (**f**). It reveals a dose-dependent decrease in translation, indicating that RG4 is essential for translation enhancement. For (**b**, **f** and **g**), data are presented as mean ± SEM of independent experiments (*n* = 3). **P < 0.005, ***P < 0.0005 and NS nonsignificant (two-sided paired *t* test). Exact mean values are provided in Supplementary Tables 3.8 and 3.9. Raw data points are provided as a Source Data file.

an RNA helicase, DHX36 (or RHAU) to unfold the RG4 structure during the translation reaction (Fig. 6d). DHX36 is a well-studied RG4-specific helicase which should effectively remove the RG4 structure formed in mRNA[48–52]. Previously, we reported an ATP-dependent repetitive unwinding mechanism by which DHX36 unfolds RG4 using single-

molecule FRET[53,54]. In agreement with our previous finding, DHX36 displayed strong affinity to RG4 and unfolded the structure even at sub-nanomolar concentration (Supplementary Fig. 4). As expected, when applied to the translation reaction, DHX36 reduced translation in a dose-dependent manner (Fig. 6e) without impacting the

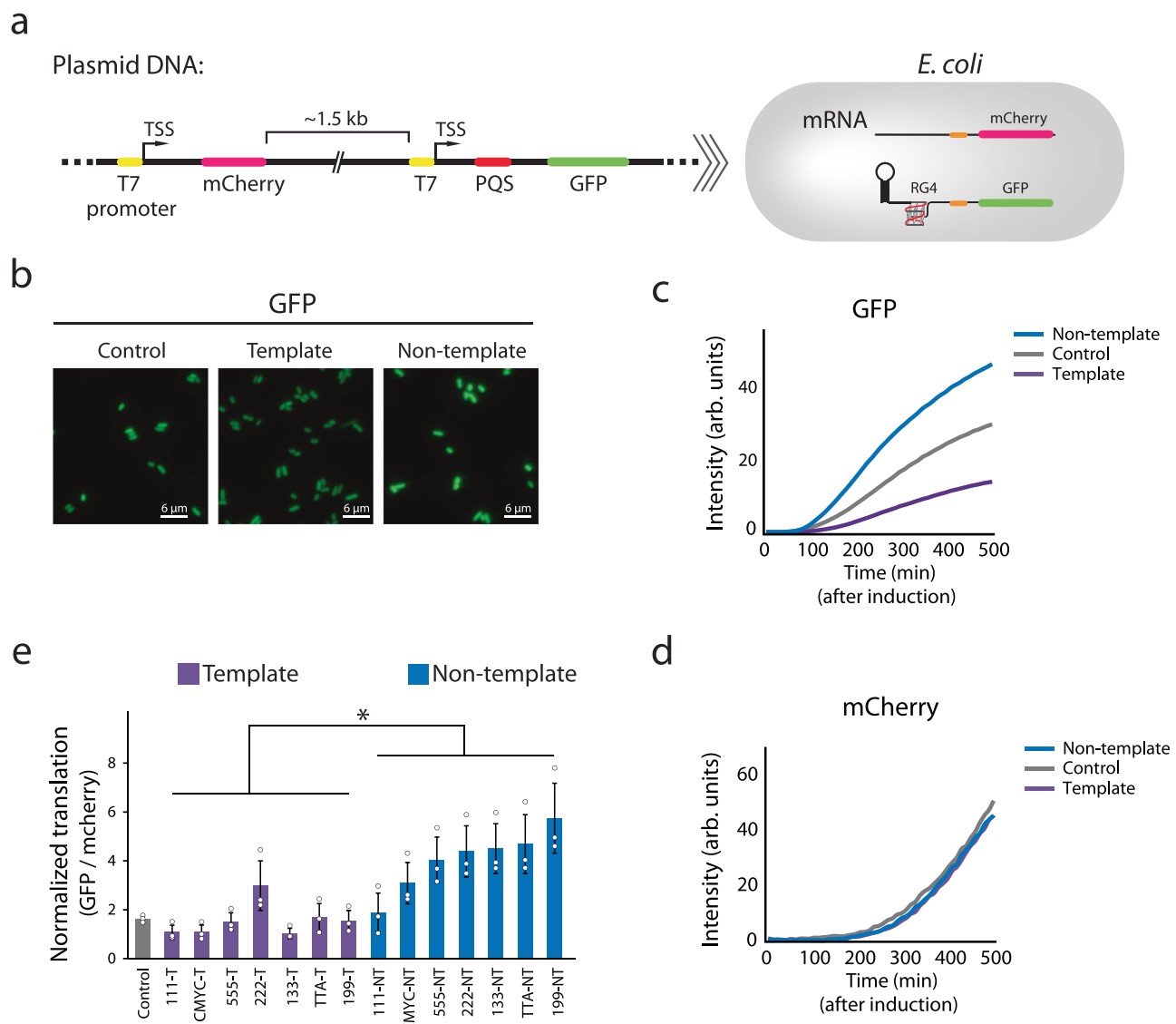

**Fig. 7 | Translation enhancement pattern is observed in *E. coli*. a** Schematic of the dual-color fluorescence reporter system. The two reporters, mCherry and GFP, have the same T7 promoter (yellow) and ribosome binding site (RBS in orange). The 5′UTR of the GFP contains the 10 hp hairpin and potential G-quadruplex sequence (PQS in red) that generates RNA G-quadruplex (RG4). **b** Fluorescence imaging of GFP expression in *E. coli*. The scale bar indicates 6 μm (50 px). The images shown are one representative result from *n* = 3 independent experiments. **c**, **d** Real-time GFP and mCherry intensities. The curves represent example traces of cMyc, where non-template (NT), control (C), and template (T) are colored in blue, gray, and purple, respectively. The data is collected after IPTG induction by plate reader. The curves shown are one representative result from *n* = 3 independent experiments. **c** The orientation-dependence is observed in GFP. **d** The three curves show no difference in mCherry. **e** Normalized translation efficiencies are defined by the ratio of GFP and mCherry signal at 210 min after induction. The PQS-NT constructs showed higher translation than T and the translation increased with bulky PQS. The calculation is described in "Method" and Supplementary Fig. 5. Data are presented as mean ± SEM of *n* = 3 independent experiments. Shown in (**e**) only represents the significance between template and non-template, where *$P < 0.05$ (two-sided paired t-test). Exact mean values are provided in Supplementary Table 3.10. Raw data points are provided as a Source Data file.

transcription (Fig. 6f, g), strongly suggesting that RG4 structure is responsible for the increased translation. We propose that the RG4 acts as a physical blockade by biasing the ribosome to move toward the translation start site.

**Translation enhancement pattern is observed in *E. coli***

Next, we asked if the hairpin and RG4-mediated translation enhancement also operates in *E. coli*. Unlike the cell-free translation system which only contains essential reagents for transcription and translation, cellular environment is enriched with other proteins, including helicases, RNA binding proteins and RNases that can modulate the gene expression process and thus change the translation enhancement effect by the 5′UTR structures[55,56]. To test this, we prepared a

dual-color fluorescence plasmid reporter. The T7 promoter-GFP was built with the 5′UTR structures for an experimental readout whereas the T7 promoter-mCherry was constructed without 5′UTR elements to serve as an internal control (Fig. 7a). The GFP expression was normalized against the mCherry expression to obtain the relative translation yield for various 5′UTR sequences (Supplementary Fig. 5a–c). The GFP expression was confirmed and visualized by fluorescence imaging (Fig. 7b). Later, we quantified the GFP and mCherry expression by acquiring real-time fluorescence and absorbance (A₆₀₀) which were recorded simultaneously by the plate reader. The GFP intensity displayed the same orientation dependence of NT > C > T as we observed in vitro (Fig. 7c), while the mCherry intensity remained similar (Fig. 7d), suggesting that the same RG4 dependent translation enhancement

occurs in *E. coli* cells. By using RT-PCR analysis, we confirmed that the difference is not based on the mRNA expression (Supplementary Fig. 5d). We cloned seven sets of plasmids with varying PQS sequences inserted either in NT or T and quantified the translation efficiency. The result reflects the same pattern as before; PQS-NT produces higher GFP signal than PQS-T and the translation further increases when bulkier PQS is inserted in NT (Fig. 7e). Taken together, our data indicates that the 5′UTR structure-dependent translational enhancement exists both in vitro and in cells.

## Discussion

Here, we applied a real-time transcription-translation coupled assay (Fig. 1) to demonstrate the impact of RG4 at 5′UTR in promoting translation (Fig. 2). This enhancement is highly correlated with the total loop length i.e., the size of the RG4 structure (Fig. 3), and this effect is further accentuated when a hairpin structure is added in tandem (Fig. 4). We demonstrate that the RG4-mediated translation enhancement is not due to elevated affinity to ribosomes, increased accessibility of RBS (Fig. 5), or improved stability of the mRNA (Fig. 6 top). The RG4 structure is the key to promoting the translation as the helicase-induced unwinding completely abolished the effect (Fig. 6 bottom). We propose that RG4 serves as a physical blockade that biases the ribosome movement thereby directing it toward protein synthesis. In addition, we demonstrate that the same mechanism operates in *E. coli* cells (Fig. 7), suggesting its potential application for controllable gene expression in *E. coli*.

We find that RG4 with longer loops induces higher translational enhancement (Fig. 3). While longer loops can enlarge the overall size of the RG4, they can weaken the folded state of the G4, based on the studies done for DNA G4 i.e., longer loop lengths lead to less stable folding of DNA-G4 due to lower folding energy[42,44]. We tested whether the folded state of RG4 varies between the short loop (111) and the longer loop (199) RG4 via smFRET assay. Surprisingly, both RNAs showed a steady high FRET state, indicating a stably folded G4 structure (Supplementary Fig. 1e, f). Furthermore, the G4 signature in the NMR spectra for cMyc-NT and 199-NT RNA verified the RG4 folding. The spectra also indicates that the presence of upstream hairpin structure may promote stable folding of RG4 (Supplementary Fig. 2a). Similarly, varying size of RG4 stalled the extension of reverse transcriptase, indicating a relatively strong stability of all the RG4 than DNA G4 we reported before by DNA polymerase stop assay[25] (Supplementary Fig. 2b). These results suggest that the G4 folding in RNA is inherently more stable than the G4 in DNA. To further weaken the RG4, we mutated the guanines in the second and third quartet to adenine to destabilize the core of RG4. Interestingly, the mutation constructs still showed a similar enhancement effect as the original cMyc RG4 (Supplementary Fig. 6), reflecting that the mutated RNA can still fold into a structure that can increase translational output. Taken together with the dual-hairpin result (Supplementary Fig. 2), translational enhancement up to 4–6 folds increase occurs regardless of the structure, but the strongest effect of over tenfold enhancement requires both stable folding and large size of the structures, for example, 177, 555, and 199 with a hairpin. In other words, our data suggests that the bulkiness of the upstream structure is more essential than the type of structure.

The effect of RG4 depends on its location as indicated in Holder and Hartig's previous work demonstrated that inserting G-quadruplex 20 bp upstream of the start codon decreased translation efficiency without changing transcription, The close proximity of G4 to the start codon likely inhibits the interaction of the 16S ribosomal RNA with the SD sequence[23,24]. Therefore, we inserted G4 at 46 bp upstream to SD sequence to prevent such structural inhibition. Whether the level of enhancement relies on the distance between RG4 and RBS, or RG4 and other structures warrants future study.

Our observation may be partially explained by the "standby model"[57,58], in which an upstream hairpin provides a temporary position for the ribosomal 30S subunit to stay, waiting for the unfolding of the SD sequence. According to this model, the rate-determining step becomes the recruitment of ribosome subunit, which is slower than the waiting time for unfolding. In our case however, the extended sequence adjacent to SD sequence (position 3 in Fig. 4a) is predicted to have a folding energy of −5.6 kcal/mol, which falls within the energy proposed by the standby model (less than −10 kcal/mol)[57]. Nevertheless, the current version of the standby model only accounts for a hairpin structure near the SD sequence, which cannot be extended to the effect of the RG4 structure positioned far from the RBS. Therefore, we propose that the RG4 structure with or without the hairpin may play the role of a blockade that biases ribosome movement.

We also considered that the translation enhancement may result from the coupled transcription and translation in *E. coli* where RNA synthesis by RNAP facilitates the recruitment of ribosomal subunits and initiates translation before the transcription is terminated i.e., co-transcriptional translation (CTT)[59,60]. However, in our system, CTT is unlikely because T7 RNAP transcription rate (220 - 230 nt/s)[61,62] is significantly higher than *E. coli* ribosome translation rate (42-51 nt/s)[63]. That is, ribosomes will be loaded onto a nascent RNA post RNA synthesis rather than being coupled to transcription. In order to examine the effect in the presence of CTT, we also cloned the same 5′ UTR sequence and GFP gene to an *E. coli* promoter $P_{L-LacO}$ system. Surprisingly, we still observed a clear increase of GFP signal in non-template construct than in template (Supplementary Fig. 7), suggesting the structural effect may still function in regular *E. coli* gene under CTT. However, the mechanism in a pure *E. coli* system should be studied more systematically in the future.

To summarize, we have demonstrated a size-dependent 5′UTR structure effect in translation enhancement in T7 and *E. coli* system (Supplementary Fig. 6d). Despite the depletion of endogenous 5′UTR RG4, we found that additional introduction of RNA G-quadruplex and hairpin structures at 5′UTR can alter the translation efficiency in the *E. coli* system. Although we examined varying sizes of both the RG4 and hairpin, our study opens a wide window of opportunity for future studies, for example, investigating the role of a stable and bulky structure, such as pseudoknot, positioned either upstream or downstream of RG4 or in between RG4 and RBS. Furthermore, we want to note that in all our experiments, PQS insertion into template (PQS-T) which produces C-rich transcript, strongly suppressed the translation efficiency both in vitro and in vivo experiments. This suggests an opposite function of C-rich RNA in down-regulating the gene expression. In conclusion, our study provides a new mechanism and function of 5′UTR mRNA in bacterial translation and provides another cloning scheme for tunable gene expression system.

## Methods

### DNA preparation

All the DNA samples were PCR-amplified from a homemade plasmid with a *sf*GFP reporter. The plasmid was derived from a GFP-expression plasmid, pZEMB8, by replacing the upstream region with T7 promoter and target 5′UTR sequence (listed in Supplementary Table 1)[25]. Additional PQS DNA were inserted into 5′UTR by restriction enzymes, EcoRI and AgeI (NEB). The recombinant plasmids were transformed into NEB-*5α* for DNA extraction and BL21-DE3 for *E. coli* expression assay. For dual-color reporter system, a mCherry gene was cloned from pET mCherry vector (Addgene, plasmid #29722) into the GFP plasmid by NEB HiFi DNA Assembly kit. Linear DNA samples for in vitro translation were PCR-amplified from GFP plasmid. The T7 promoter forward primer and T7 terminator reverse primers (see Supplementary Table 2) were designed and ordered from Integrated DNA Technologies (IDT) for amplification. The amplified linear DNA held the T7 promoter, 5′ UTR with PQS, GFP gene, and T7 terminator and was purified by gel electrophoresis and Gel Extraction Kit (QIAquick).

## RNA preparation

RNA samples were prepared by HiScribe T7 Quick High Yield RNA Synthesis kit (NEB) at 37 °C overnight. Each reaction (20 μL) had 1 μg of DNA, NTP mix, T7 RNA Polymerase, and RNase-free water. The overnight product was firstly digested by DNase I (0.1 U/μL) in DNase reaction buffer (10 mM Tris-HCl pH 7.6, 2.5 mM $MgCl_2$, 0.5 mM $CaCl_2$) at 37 °C for 30 min. Later, the reaction was quenched by adding 1 μL 0.5 M EDTA, followed by inactivating at 75 °C for 10 min. RNA was purified by Monarch RNA CleanUp kit (NEB).

## In vitro translation assay

Ensemble in vitro translation assay was conducted by PURExpress In Vitro Translation kit (NEB) and performed by TECAN Spart plate reader at 37 °C. Each reaction (25 μL) was premixed with 55 ng linear DNA (4 nM), 400 nM molecular beacon (see Supplementary Table 2), RNase inhibitor murine (0.8 unit/μL), and 10 μL kit solution A. To measure the RG4 formation, N-methyl mesoporphyrin IX (NMM) was added in premixed solution at final concentration of 1 μM. The reaction was initiated by adding 7.5 μL kit solution B and loaded on a 384-well plate (white and transparent bottom, Thermo Scientific). The Cy3, GFP, and NMM were excited at $\lambda_{ex}$ 545, 485, and 393 nm and detected at $\lambda_{em}$ 570, 510, and 610 nm, respectively. Both excitation and emission were assigned with 10 nm slit size. The initiation transcription rate was quantified from the linear part (10–25 min) of the Cy3 intensity curve. The translation rate was quantified from the linear part (25–50 min) of the GFP intensity curve. Each rate was normalized to the transcription rate and translation rate of 10 hp control DNA construct, respectively. The normalized translation efficiency was calculated by dividing the normalized translation rate to the normalized transcription rate. The half-life was defined by the time that the intensity reached 50% of the plateau.

## In vitro transcription NMM assay

Ensemble in vitro transcription for real-time N-methyl mesoporphyrin IX (NMM) measurement was performed by TECAN Spark plate reader at 37 °C. Each sample was prepared with 1 nM linear DNA template in transcription buffer (40 mM Tris-HCl pH 8.3, 50 mM KCl, 6 mM $MgCl_2$, 2 mM spermidine, 1 mM dithiothreitol), RNase inhibitor murine (0.4 unit/μL), T7 RNA polymerase (1.25 unit/μL), and 1 mM NMM. The reaction was initiated by adding an NTP mix for a final concentration of 1 mM. Each reaction (100 μL) was loaded on 96-well transparent plate (Thermo Scientific). The data was collected at $\lambda_{ex}$ 393 nm (slit size 10 nm) and $\lambda_{em}$ 610 nm (slit size 10 nm). For emission spectrum, the data were collected at $\lambda_{ex}$ 393 nm (slit size 10 nm) and $\lambda_{em}$ 580–650 nm (slit size 10 nm).

## RNA G4 competition assay

RNA competition assay was modified from in vitro translation assay by adding RNA competitors. PolyU 40 and cMyc RG4 were ordered from IDT. Other RNAs were synthesized from the 5′UTR of 10 hp control and 10 hp cMyc DNA. The DNA templates were PCR-amplified by T7 promoter primer, short-length primer 1, and long-length primer 2 (see Supplementary Table 2). RNA purification protocol was described in *RNA Preparation*. The reaction was performed with 55 ng of 10 hp cMyc-NT DNA and 5 μM of competitor RNA by plate reader at 37 °C.

## RBS probe binding assay and accessibility assay

A molecular beacon is complementary to the ribosome binding site (RBS) was ordered from IDT and labeled by Cy5 and quencher at each end (see Supplementary Table 2). The Cy5 intensity of beacon was measured at $\lambda_{ex}$ 640 nm and $\lambda_{em}$ 665 nm with 10 nm slit size. The binding assay was conducted by incubating 400 nM beacon and RNAs of 0 hp control, T, and NT at 37 °C. RNA purification protocol was described in *RNA Preparation*. The data points were collected by titrating RNA concentration in a twofold series dilution from 1.75 μM

to 24 nM. The $K_d$ was fitted to the binding curve by OriginPro. For accessibility assay, each sample (10 μg of RNA) was mixed with 400 nM of the RBS beacon and incubated with or without heating treatment. The heated samples were incubated at 80 °C for 5 min and cooled by 10 °C intervals for 5 min until 37 °C. After incubation, the intensities of all the samples were measured by TECAN plate reader at 37 °C.

## RT-qPCR

For in vitro translation, post-translation samples (20 μL) were treated with DNase I (0.1 U/μL) in DNase reaction buffer (10 mM Tris-HCl pH 7.6, 2.5 mM $MgCl_2$, 0.5 mM $CaCl_2$) at 37 °C for 30 min. The reaction was quenched by 1 μL of 0.5 M EDTA, followed by heat inactivation at 75 °C for 10 min. The DNA-free samples were diluted tenfold, and 2 μL of each diluted sample was used to synthesize cDNA by ProtoScript II cDNA Synthesis Kit (NEB). The reaction was incubated at 25 °C for 5 min, followed by 42 °C for 1 h and 80 °C for 5 min. The cDNA was diluted 100-fold before qPCR measurement. The qPCR sample (20 μL) contained 1 μL 100-fold diluted cDNA, 10 μL SYBR Green Supermix (BioRad), and 250 nM primers (see Supplementary Table 2).

## DHX36 purification and titration experiment

The *E. coli* strand with DHX36 plasmid was made in the lab and described in a previous publication[53]. The *E. coli* was inoculated in TB medium and grew overnight. Next day, the culture was diluted to $OD_{600}$ of 0.01 and grew till $OD_{600}$ of 0.6 at 37 °C and induced protein expression with 1 mM IPTG at 14 °C overnight. The purification protocol followed previous publications[53]. Protein concentration was quantified by standard BSA (NEB) calibration curve by SDS-PAGE, and the aliquots of protein samples were stored at −80 °C. For DHX36 titration assay, the protein was diluted by TNM buffer (10 mM Tris-HCl, pH 8.0; 50 mM NaCl; 5 mM $MgCl_2$) to avoid the change of reaction volume and buffer condition. 1 μL of each titrated DHX36 and additional 1 mM ATP were premixed with kit solution A, and the reaction was initiated by adding kit solution B. The experimental protocol is described in *the In Vitro Translation Assay*.

## *E. coli* dual fluorescence assay

PQS-contained dual fluorescence plasmids, described in *DNA Preparation*, were transformed into BL21(DE3) *E. coli*, and grew in LB medium at 37 °C overnight. The cultures were diluted to $OD_{600}$ of 0.01 and grew at 37 °C by TECAN plate reader with a 24-well transparent plate (1 mL for each culture). $OD_{600}$ measurements were performed every 10 min, followed by an orbital shaking mode (215 rpm) for a 10 min interval. The gene expression was induced by 1 mM IPTG at $OD_{600}$ of 0.4. After the induction, the protein expression was monitored by an auto-loop measurement of 10 min shaking, $OD_{600}$, GFP ($\lambda_{ex}$ 485/$\lambda_{em}$ 510, slit 10 nm), and mCherry ($\lambda_{ex}$ 585/$\lambda_{em}$ 610, slit 10 nm) for 10 h. The translation efficiency was defined by the ratio of GFP to mCherry and normalized to real-time $OD_{600}$ (Supplementary Fig. 5). The ratio at 210 min after induction represented the maximal efficiency of each strain. The cultures after 210-min induction were collected and extracted mRNA for RT-qPCR. The pellet was treated with 20 mg/mL lysozyme, and the mRNA was extracted by using Qiagen RNeasy Kits. The primers of qPCR are listed in Supplementary Table 2.

## smFRET assay

The PQS RNA oligonucleotides (see Supplementary Table 2) for smFRET were purchased from IDT with terminal amine modification for Cy3 labeling. The 18-mer RNA primer for immobilizing PQS RNA on slides was purchased from IDT and later labeled Cy5. The labeling protocol was described in a previous publication. RNA samples were annealed in TE buffer (10 mM Tris-HCl and 1 mM EDTA, pH 8) at the ratio 1:1. The mixtures were heated at 80 °C for 5 min and slowly

cooled to room temperature (1 °C per minute). The single molecule assays were performed by using a home-built prism-type total internal reflection fluorescence microscope (TIRFM) at room temperature (23.0 ± 1.0 °C)[25,64]. RNA sample (10 nM) was diluted to 25 pM and immobilized on a PEG-coated quartz slide by neutravidin (0.05 mg/mL). The reaction buffer (50 mM Tris-HCl, pH 7.5; 50 mM NaCl; 50 mM KCl; 5 mM $MgCl_2$; 5% glycerol; 80 units RNase inhibitor murine) with an oxygen scavenging system (1 mg/mL glucose oxidase, 0.8% v/v glucose, ~10 mM Trolox, and 0.03 mg/mL catalase). The smFRET experiment was performed by two solid-state lasers, 532 nm and 640 nm lasers. Each measurement was recorded with a 100 ms time resolution by smCamera software and analyzed with Interactive Data Language (IDL). The outputs were processed with custom MATLAB script to generate trajectories and FRET histograms[64,65]. For DHX36 assay, protein (1 nM) or ATP (1 mM) was premixed into the reaction buffer, and a flow system was applied to study real-time binding and unwinding events[65].

### Reverse transcriptase stop assay

RNA samples were prepared from the 5′UTR of 0 hp control, cMyc-T, and NT of 111, cMyc, 133, 222, TTA, 444, 199 DNA. The DNA templates were PCR-amplified by T7 forward primer and long-length primer 2 (see Supplementary Table 2). RNA synthesis and purification were described in *RNA Preparation*. A Cy3-labeled long-length primer 2 was used to visualize the extended cDNA product. The primer was premixed in reaction buffer (50 mM Tris-HCl, pH 8.3, 75 mM KCl, 3 mM $MgCl_2$, 10 mM DTT, RNase inhibitor murine (0.4 unit/μL), and 0.5 mM dNTP. RNA was added into primer-mixed reaction buffer at final concentration of 100 nM RNA and 500 nM primer. The samples were annealed with a stepwise heating and cooling process, 70 °C for 5 min, 25 °C for 5 min, and −80 °C for 10 min). After annealing, reverse transcriptase M-MuLV (NEB) was added (10 unit/μL) into the sample, and the reaction was incubated at 42 °C for 1 h. Then, the samples were added with 2.5 unit RNase H (NEB) and 5 μg RNase A (Thermo Scientific) and incubated at 37 °C for 15 min to remove the RNA. Finally, the whole reaction was stopped by adding 0.5 μL of 0.5 M EDTA. The extended cDNA was distinguished by 6% TBE-urea gel, run at constant 230 V for 30 min. The image was taken by gel imager (Amersham imager 600) with 520 nm LED light.

### NMR assay

RNA samples were prepared from the 5′UTR of 0 hp cMyc-NT, 10 hp cMyc-NT, 0 hp 199-NT, and 10 hp 199-NT DNA. The DNA templates were PCR-amplified by T7 forward primer and short-length primer 1 (see Supplementary Table 2). RNA synthesis and purification protocol was described in *RNA Preparation*. The length of 0 hp and 10 hp cMyc-NT RNA was 94 bases, and the length of 0 hp and 10 hp 199-NT RNA was 109 bases. For NMR, the samples contained 90–130 μM RNA in 20 mM potassium phosphate buffer pH 7.0, and 5% $D_2O$. The cMyc-NT samples contained 70 mM KCl, and the 199-NT samples contained 150 mM KCl. The final sample volume was adjusted to 130 μL and loaded into a 3 mm Wilmad tube (Sigma Aldrich). 1D $^1H$ spectra were obtained on an 800 MHz ($^1H$) Bruker NEO spectrometer, equipped with a triple-resonance cryogenic probe. A double-echo watergated sequence zggpw5 from the Bruker library was used for high-quality water suppression and was optimized for maximum excitation in the 10–16 ppm region. Experimental parameters were as follows: 60 °C temperature, 80 ms acquisition time and 512 scans per FID; interscan delay 1.5 s; interpulse delay in the zggpw5 sequence 30 ms; total time per spectrum 14 min.

### Statistics analysis

Data shown in Figs. 2–7 were obtained from individual and independent experiments. All the numbers were calculated and presented in value ± SEM. The statistics tests were calculated by two-sided paired or unpaired *t* test, depending on the data. The average numbers, SEM, and statistics *P* values were reported in Supplementary Table 3. The raw data points were provided in a Source Data file.

### Reporting summary

Further information on research design is available in the Nature Portfolio Reporting Summary linked to this article.

## Data availability

The data supporting the findings of this study are available from the corresponding authors upon request. Source data for the figures and Supplementary Figs. are provided as a Source Data file. Source data are provided with this paper.

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

## Acknowledgements

We thank all the members of the Sua Myong and Taekjip Ha laboratories for their helpful comments and constructive criticisms. We appreciate Ananya Majumdar and JHU BioNMR Center for NMR technique support. This work was supported by 1R01GM149729-01.

## Author contributions

C.Y.L. and S.M. designed experiments. C.Y.L., M.J., and A.W. conducted experiments. C.Y.L., M.J., and S.M. wrote the paper.

## Competing interests

The authors declare no competing interests.
