## [Peer Review File · Nature Communications]

5'UTR G-quadruplex structure enhances translation in size dependent mannerREVIEWER COMMENTS

Reviewer #1 (Remarks to the Author):

In this study, Lee et al. exploit a dual fluorescence system to monitor translation (fluorescence of super-folder GFP) versus transcription (base pairing of a DNA molecular beacon) for a variety of constructs with different G-quadruplex (G4) sequences upstream of the ribosome binding sites. Using these constructs driven by the T7 promoter in PURExpress In vitro translation reactions or E. coli cells, the authors report that G4 sequences increase translation, particularly if the G4 loops are long and if the G4 sequence is preceded by a stem loop. After showing that the increase is not due to increased ribosome affinity, increased ribosome binding site accessibility or increased mRNA stability, the authors conclude that the structures prevent ribosome dislodging.

The manuscript is clearly written, and the findings are straightforward and well documented.

However, importantly, it is unclear whether the findings have any physiological relevance. The study would be a much more meaningful contribution if the authors documented an endogenous gene with a G4 sequence where the presence of the G4 structure impacts expression of the downstream gene. There are standard assays that have been carried out to examine regulation by riboswitches, uORFs, or small, regulatory RNAs in 5' UTR sequences. Additional evidence for the physiological relevance of a G4 could come from conservation of the element upstream of the same gene across multiple species. Even in this case there is the question whether the translation of the corresponding mRNA is constitutively increased by the G4 element or whether there are conditions where formation of the element is regulated.

Reviewer #2 (Remarks to the Author):

This is an exceptionally good manuscript on the role of G4 in translation. The data is solid, however, the model proposed in the abstract doesn't make sense. The authors model states that failure to dislodge ribosomes would result in increased translation. However, numerous pieces of literature says that failure to dislodge is repressive. This should be corrected. The authors see activation upon insertion of G4 into a reporter so, therefore, the model must include activation. One could assume direct recognition of the G4 by the ribosome, facilitating recruitment. But, it can't be repressive and activating at the same time.

Reviewer #3 (Remarks to the Author):

This paper reports the discovery that a bulky structure in the 5' untranslated region of an mRNA increases translation efficiency. The work is a continuation of the group's earlier work on such structures' impacts on transcription. G-quadruplex presence in the mRNA is the primary structure of investigation, but hairpins are also examined. The finding is that increasing bulkiness of rna-Gquadruplexes correlates with increased translation efficiency and there is a further increase induced by including hairpins (although hairpins alone are not enough). Most of the work uses cell-free translation systems although an experiment in E. coli is sufficiently convincing to imply the phenomena is also present in live cells.

Several experiments are performed to address possible mechanisms. Convincing evidence is presented that the translation efficiency enhancement is not caused by increased recruitment of ribosomes to the mRNA or increased mRNA stability. The authors propose the effect derives from the steric block the bulky structure provides to prevent ribosome dissociation (implied by sliding off the end). This is a very intriguing and reasonable idea. Perhaps further study is required to address this possibility in

more detail, but I don't consider that to be required as this current manuscript is a coherent and interesting story.

I do not find problems with this paper. The effect is large, proper controls are included, the tests to rule out several hypotheses are convincing. The paper is very clearly written, experiments appear expertly performed, and all required methods are described. I support publication.

I offer a couple small points the authors might consider, but overall I found this a clear, interesting and convincing piece of work.

Questions:

Is there evidence that the binding of the molecular beacon to the RNA does not impact the rates of translation? Such evidence might be in previous publications, but specific mention of this fact would strengthen the conclusions of this paper.

The increased translation rate in 2e is described in the main text as normalized to the transcription rate, thus showing increases beyond those expected due to the increased transcription. This is quite significant and could be emphasized more in the figure/figure caption for those readers who might look only at the figures and not give as much time to the main text.

Very Minor Points:

In the results section "RNA G4 increases translation efficiency", 2nd paragraph, sentence "Surprisingly, the translation reporter... induced over four-fold higher protein product compared to the control" seems redundant with the Results section "Bulkiness of RG4 drives translational enhancements, sentence "Surprisingly, PQS-NT sequences led to 4-fold higher translation than the control"

Second to last sentence of section "RG4 may stabilize ribosome bound to mRNA", the authors say "Strikingly, when applied to the translation reaction, DHX36 reduced translation ... without impacting transcription". It seems this is expected because DHX36 removes RG4 and the RG4 stimulated translation. It could not be any other way. Maybe not "Strikingly"? Maybe, "as expected"?

Reviewer #4 (Remarks to the Author):

The manuscript by Lee et al. proposes that RNA G-quadruplexes (G4) increase the rate of translation in *E. coli* by preventing ribosome dislodgement. It begins with the observation that sequences that can fold into G4 RNA stimulate translation in co-transcriptional translation assays in vitro. These effects become more pronounced as G4 loop size increases. Next, the authors tested 4 hypotheses that could explain these effects: (i) increasing ribosome affinity, (ii) increasing RNA accessibility, (iii) stabilizing RNA or (iv) stabilizing ribosomes on RNA. Finally, they propose in vivo assays using plasmids with sequences predicted to fold into G4s and show that they also promote translation in *E. coli*.

In general, the topic and questions are of great relevance to the field, but while the effects of G-rich sequences on translation are very convincing, the roles of G-quadruplex structures are not demonstrated. On the contrary (particularly in Fig. 3 and Supplementary Fig. 5), the data support that G4 structures are not involved in translation enhancement. Consequently, the manuscript does not contain sufficient evidence for the effects of G4 structures on translation to be published in *Nature Communications*.

Main comments:

1- What sequences are used in Fig.2 (scrambled control sequence (C) and G4 RNA (NT))? (Are these the 111 and control sequences shown in Fig.3?)

To show that G-quadruplexes are involved in increased translation, it is necessary to test mutations that affect G-quadruplex formation. This has been done in Supplementary Fig.5. Actually, here the data sets show that G-quadruplex formation is not necessary for this phenomenon, since all mutants (M1-4) can increase translation (Supplementary Fig. 5.).

While authors mention "mutated RNA can still fold into a structure that can increase translational output", these are not G-quadruplexes !

Fig.1 and supplementary Fig.5 should be merged.

Additional control sequences retaining GC content but unable to fold into G4 (this can be tested using Quadron, G4mismatch, G4hunter...) should have been tested. Despite all efforts of the authors in this work, these controls show that effects on translation do not rely on G-quadruplex formation. Authors could test if G-richness, GC content, ... are important. It is intriguing to observe that why G-rich sequences enhance translation (NT), C-rich sequences (T) inhibit it when compared to the control sequence (Fig.3).

2- While the effects of loop size on translation are clear (Fig.3), constructs that show the least potential and least stable G-quadruplexes have the strongest effects. This again object to the role of G4 in this process. In this assay, the most stable G-quadruplexes (111, cMyc and 222) show the same translational efficiency as the control sequence. The authors should conclude from this series of experiments that the formation of G-quadruplex is not involved in translation enhancement.

3- The authors must show in fig.2a and fig.3 the presence and absence of hairpin. Fig4.b should also show the control without hairpin.

4- Effects on mRNA stability measured by qPCR should be represented as RNA quantity using standard qPCR quantification methods. As presented, this experiment could not be interpreted. I can see at least 1 or 2 cycles of difference between the curves (i.e. about 2 to 4 fold changes!). As a control, the amount of RNA at T0 should also be quantified by qPCR.

5- The DHX36 experiment is very elegant, but it is very difficult to distinguish between effects of DHX36 binding or G4 unwinding. The authors should have tested DHX36 without ATP.

6- The authors tested the in vivo effects of G4-putative sequences on translation, they should also have tested series of E. coli 5'UTR sequences containing RG4 and controls with mutations altering the G4 structures. However, previous in vitro experiments show that G4 structures are not involved. Controls of NT sequences with mutations altering G4 formation are also missing.

Other comments

a- The graphical summary of the methodology (Fig.1a) should indicate the detection methods (i.e. Beacon for transcription; GFP for translation).

b- Supplementary fig.1d, Intensity and FRET tracks of a control sequence that cannot fold into G4 should also be shown.

c- Supplementary fig.2b CMNT should be called CMYC.

d- Fig.3b needs sequence labels.

e- Supplementary fig. 1b and Supplementary fig.4a-c need legends.

f- Is the difference between competitors 5 and 6 on translation significant (Fig.5d)?

g- The effects of the G4 structure could have been tested in assays with K⁺ or Li⁺ ionic conditions that stabilize and destabilize G4 formation respectively.

POINT BY POINT RESPONSE TO REVIEWERS' COMMENTS

(responses in blue, changed text in red)

Reviewer #1 (Remarks to the Author):

In this study, Lee et al. exploit a dual fluorescence system to monitor translation (fluorescence of super-folder GFP) versus transcription (base pairing of a DNA molecular beacon) for a variety of constructs with different G-quadruplex (G4) sequences upstream of the ribosome binding sites. Using these constructs driven by the T7 promoter in PURExpress In vitro translation reactions or *E. coli* cells, the authors report that G4 sequences increase translation, particularly if the G4 loops are long and if the G4 sequence is preceded by a stem loop. After showing that the increase is not due to increased ribosome affinity, increased ribosome binding site accessibility or increased mRNA stability, the authors conclude that the structures prevent ribosome dislodging.

The manuscript is clearly written, and the findings are straightforward and well documented.

However, importantly, it is unclear whether the findings have any physiological relevance. The study would be a much more meaningful contribution if the authors documented an endogenous gene with a G4 sequence where the presence of the G4 structure impacts expression of the downstream gene. There are standard assays that have been carried out to examine regulation by riboswitches, uORFs, or small, regulatory RNAs in 5' UTR sequences. Additional evidence for the physiological relevance of a G4 could come from conservation of the element upstream of the same gene across multiple species. Even in this case there is the question whether the translation of the corresponding mRNA is constitutively increased by the G4 element or whether there are conditions where formation of the element is regulated.

- We acknowledge the reviewer's comment on the physiological relevance of increased translation efficiency due to the 5'UTR G4 structure. Studies done on endogenous mammalian G4 indicate that G4 formation usually results in translation inhibition (Kumari, Sunita, *et al.* (2007), Arora, Amit *et al.* (2008), Agarwala, Prachi *et al.* (2014, 2019)). However, the translation control mechanism in mammalian cells differs from *E. coli*. Genome-wide analysis by Rawal P *et al.* (2006) across 18 different prokaryotes has shown that DNA G4 motifs are enriched in promoter regions of orthologous genes across the prokaryotes. Recent analysis by Kaplan OI *et al.* (2016) shows two conserved G4 motifs in *E. coli* mainly distributed in intergenic regions. Experimentally, a study by Shao, X. *et al.* (2020) demonstrated that endogenous RNA G4 predominantly forms at coding region and can both up- or downregulate translation across different prokaryotes. Furthermore, the effect of G4 forming sequence on gene expression has been confirmed as a combined dependence of inserted position and orientation across promoter to start codon (Holder IT, Hartig JS (2014)), revealing a potential mechanism of translation regulation by G4.

Endogenous G4 sequences in *E. coli* differ in G-tracts and loop sequences, leading to various potential G4-forming sequences and structures. As our study aims to conduct a biophysical

characterization of the effect of G4 structure on translation enhancement, we used G4 forming sequences that have been characterized before in our lab (2014, 2015, 2016, 2019 denoted by * in references below). Additionally, we maintained the location of the G4 forming sequence, and the sequence downstream of the G4 to eliminate sequence context effects in *E. coli*. Future directions include inserting G4-forming sequences into the *E. coli* genome to observe the endogenous effect of the structure.

While riboswitches, uORFs, and small RNA are well-studied motifs that typically work, though modulating access to the ribosome binding site or start codon (Wassarman MK (2002), Abduljalil JM (2018), Chen, Fan *et al.* (2022)), the mechanisms through which G4 structures can affect *E. coli* translation are unknown. We used GFP fluorescence detection instead of the common β -galactosidase activity assay to determine translation regulation due to the higher efficiency, reproducibility, and throughput of fluorescent measurements. One major take-away of our study is for the broad research community to utilize our G4-hairpin constructs for tunable expression of their protein of interest.

- We inserted the following paragraph in the revised manuscript to address the point about the endogenous 5'UTR RG4.

"In addition, despite the low abundance of PQS in prokaryotes, computational studies have identified a few conserved G4 motifs across prokaryotic species positioned non-randomly in promoter and intergenic region, indicating an evolutionarily conserved function of G4 in bacterial genome^{35, 36, 37}. A recent transcriptome analysis finds that most folded RG4 in bacteria are two-quartet G quadruplex and mainly detected in the coding region and underlines the function of RG4 up- or down-regulating gene expression differs between species³⁸."

References:

- Kumari S, Bugaut A, Huppert JL, Balasubramanian S. An RNA G-quadruplex in the 5' UTR of the NRAS proto-oncogene modulates translation. *Nat Chem Biol* **3**, 218-221 (2007).
- Arora A, Dutkiewicz M, Scaria V, Hariharan M, Maiti S, Kurreck J. Inhibition of translation in living eukaryotic cells by an RNA G-quadruplex motif. *RNA* **14**, 1290-1296 (2008).
- Agarwala P, Pandey S, Maiti S. Role of G-quadruplex located at 5' end of mRNAs. *Biochim Biophys Acta* **1840**, 3503-3510 (2014).
- Agarwala P, Pandey S, Ekka MK, Chakraborty D, Maiti S. Combinatorial role of two G-quadruplexes in 5' UTR of transforming growth factor beta2 (TGFbeta2). *Biochim Biophys Acta Gen Subj* **1863**, 129416 (2019).
- Rawal P, Kummarasetti VB, Ravindran J, Kumar N, Halder K, Sharma R, Mukerji M, Das SK, Chowdhury S. Genome-wide prediction of G4 DNA as regulatory motifs: role in Escherichia coli global regulation. *Genome Res* **16**, 644-655. (2006)
- Kaplan OI, Berber B, Hekim N, Doluca O. G-quadruplex prediction in E. coli genome reveals a conserved putative G-quadruplex-Hairpin-Duplex switch. *Nucleic Acids Res* **44**, 9083-9095 (2016).
- Holder IT, Hartig JS. A matter of location: influence of G-quadruplexes on Escherichia coli gene expression. *Chem Biol* **21**, 1511-1521 (2014).

Shao X, Zhang W, Umar MI, Wong HY, Seng Z, Xie Y, Zhang Y, Yang L, Kwok CK, Deng X. RNA G-Quadruplex Structures Mediate Gene Regulation in Bacteria. *ASM Journals* **11**, 10.1128/mbio.02926-19 (2020)

*Tippana R, Xiao W, Myong S. G-quadruplex conformation and dynamics are determined by loop length and sequence. *Nucleic Acids Res.* **42**, 8106-8114. (2014)

*Kreig A, Calvert J, Sanoica J, Cullum E, Tippana R, Myong S. G-quadruplex formation in double strand DNA probed by NMM and CV fluorescence. *Nucleic Acids Res* **43**, 7961-7970 (2015).

*Tippana R., Hwang H., Bohr V.A., Opresko P.L., Myong S., "Single molecule imaging reveals a common mechanism shared by G-quadruplex resolving helicases", *PNAS*;113(30):8448-53 (2016)

*Tippana R, Chen MC, Demeshkina NA, Ferre-D'Amare AR, Myong S. RNA G-quadruplex is resolved by repetitive and ATP-dependent mechanism of DHX36. *Nat Commun* **10**, 1855 (2019)

Wassarman MK. Small RNAs in Bacteria: Diverse Regulators of Gene Expression in Response to Environmental Changes. *Cell.* **109**, 141-144 (2002)

Abduljalil JM. Bacterial riboswitches and RNA thermometers: Nature and contributions to pathogenesis. *Noncoding RNA Res* **3**, 54-63 (2018).

Chen, F., Coccagn-Bousquet, M., Girbal, L., & Nouaille, S. 5'UTR sequences influence protein levels in *Escherichia coli* by regulating translation initiation and mRNA stability. *Frontiers in microbiology*, **13**, 1088941. (2022).

Reviewer #2 (Remarks to the Author):

This is an exceptionally good manuscript on the role of G4 in translation. The data is solid, however, the model proposed in the abstract doesn't make sense. The authors model states that failure to dislodge ribosomes would result in increased translation. However, numerous pieces of literature say that failure to dislodge is repressive. This should be corrected. The authors see activation upon insertion of G4 into a reporter so, therefore, the model must include activation. One could assume direct recognition of the G4 by the ribosome, facilitating recruitment. But, it can't be repressive and activating at the same time.

- We acknowledge a misleading sentence. We have modified the sentence to point out that the upstream structure restricts the direction of ribosome movement.

In the abstract:

"We propose a physical barrier model in which bulky structures in 5'UTR **biases ribosome movement toward downstream start codon**, thereby increasing the translation output. "

In the main text:

RG4 may be a physical blockade that biases the ribosome movement toward AUG

The negative results obtained in the first three hypotheses strongly suggest that the effect of RG4 in translational regulation must occur after the ribosome loads on the mRNA. This raises a possibility that RG4 may promote translation **by biasing the ribosome movement toward the AUG position** (Fig. 6c). To test the hypothesis, we applied an RNA helicase, DHX36 (or RHAU) to unfold the RG4 structure during the translation reaction (Fig. 6d). DHX36 is a well-studied RG4-specific helicase which should effectively remove the RG4 structure formed in mRNA^{48, 49, 50, 51, 52}. Previously, we reported an ATP-dependent repetitive unwinding mechanism by which DHX36 unfolds RG4 using single-molecule FRET^{53, 54}. In agreement with our previous finding, DHX36 displayed strong affinity to RG4 and unfolded the structure even at subnanomolar concentration (Supplementary Fig. 4). As expected, when applied to the translation reaction, DHX36 reduced translation in a dose-dependent manner (Fig. 6e) without impacting the transcription (Fig. 6f and 6g), strongly suggesting that RG4 structure is responsible for the increased translation. We propose that the RG4 acts as a physical blockade by **biasing the ribosome to move toward the translation start site**.

Reviewer #3 (Remarks to the Author):

This paper reports the discovery that a bulky structure in the 5' untranslated region of an mRNA increases translation efficiency. The work is a continuation of the group's earlier work on such structures' impacts on transcription. G-quadruplex presence in the mRNA is the primary structure of investigation, but hairpins are also examined. The finding is that increasing bulkiness of rna-Gquadruplexes correlates with increased translation efficiency and there is a further increase induced by including hairpins (although hairpins alone are not enough). Most of the work uses cell-free translation systems although an experiment in *E. coli* is sufficiently convincing to imply the phenomena is also present in live cells.

Several experiments are performed to address possible mechanisms. Convincing evidence is presented that the translation efficiency enhancement is not caused by increased recruitment of ribosomes to the mRNA or increased mRNA stability. The authors propose the effect derives from the steric block the bulky structure provides to prevent ribosome dissociation (implied by sliding off the end). This is a very intriguing and reasonable idea. Perhaps further study is required to address this possibility in more detail, but I don't consider that to be required as this current manuscript is a coherent and interesting story.

I do not find problems with this paper. The effect is large, proper controls are included, the tests to rule out several hypotheses are convincing. The paper is very clearly written, experiments appear expertly performed, and all required methods are described. I support publication.

I offer a couple small points the authors might consider, but overall I found this a clear, interesting and convincing piece of work.

➤ We thank the reviewer for the enthusiasm! Below, we addressed the reviewer's questions.

Questions:

1. Is there evidence that the binding of the molecular beacon to the RNA does not impact the rates of translation? Such evidence might be in previous publications, but specific mention of this fact would strengthen the conclusions of this paper.

➤ We performed an additional experiment to test the translation rate in the absence or presence of the molecular beacon (shown in the figure below). As shown, the translation rate does not differ between the two conditions. This newly obtained result is added as Supplementary Figure 1a in the revised manuscript.

2. The increased translation rate in 2e is described in the main text as normalized to the transcription rate, thus showing increases beyond those expected due to the increased transcription. This is quite significant and could be emphasized more in the figure/figure caption for those readers who might look only at the figures and not give as much time to the main text.

➤ We thank the reviewer for informing us. We have modified the figure caption as follows.

e_{NT} construct enhanced the translation 4-fold higher than C and T. Translation efficiencies are calculated from the translation rates obtained from the early linear part of the curve in **d** and the normalized transcription rates in **c**. The translation efficiency was normalized to the control sequence."

Very Minor Points:

In the results section "RNA G4 increases translation efficiency", 2nd paragraph, sentence "Surprisingly, the translation reporter... induced over four-fold higher protein product compared to the control" seems redundant with the Results section "Bulkiness of RG4 drives translational enhancements, sentence "Surprisingly, PQS-NT sequences led to 4-fold higher translation than the control"

➤ We apologize for the incorrect statement. In figure 3c, we emphasized the PQS-NT without upstream hairpin structure also enhance translation, but in figure 2e, the construct contains a 10bp hairpin. Here is the revised sentence.

"Surprisingly, **without upstream hairpin structure**, all the PQS-NT sequences still led to higher translation than the control and template (Fig. 3c),"

Second to last sentence of section "RG4 may stabilize ribosome bound to mRNA", the authors say "Strikingly, when applied to the translation reaction, DHX36 reduced translation ... without impacting transcription". It seems this is expected because DHX36 removes RG4 and the RG4

stimulated translation. It could not be any other way. Maybe not "Strikingly"? Maybe, "as expected"?

- We agree with the reviewer, and the sentence was modified as suggested.

Reviewer #4 (Remarks to the Author):

The manuscript by Lee et al. proposes that RNA G-quadruplexes (G4) increase the rate of translation in *E. coli* by preventing ribosome dislodgement. It begins with the observation that sequences that can fold into G4 RNA stimulate translation in co-transcriptional translation assays in vitro. These effects become more pronounced as G4 loop size increases. Next, the authors tested 4 hypotheses that could explain these effects: (i) increasing ribosome affinity, (ii) increasing RNA accessibility, (iii) stabilizing RNA or (iv) stabilizing ribosomes on RNA. Finally, they propose in vivo assays using plasmids with sequences predicted to fold into G4s and show that they also promote translation in *E. coli*.

In general, the topic and questions are of great relevance to the field, but while the effects of G-rich sequences on translation are very convincing, the roles of G-quadruplex structures are not demonstrated. On the contrary (particularly in Fig. 3 and Supplementary Fig. 5), the data support that G4 structures are not involved in translation enhancement. Consequently, the manuscript does not contain sufficient evidence for the effects of G4 structures on translation to be published in Nature Communications.

- We appreciate reviewer's critical comments. To address the criticism, we tested the role of G-rich sequences in translation by conducting new sets of experiments as summarized here and explained in detail further below.
 1. New sets of **mutants with varying G density** were tested in translation assay.
 2. **Reverse transcriptase stop** assay was performed to probe the structural impedance by the G4 structures,
 3. **NMR** was employed to show the formation of the G4 structures used in our study.

Main comments:

1- What sequences are used in Fig.2 (scrambled control sequence (C) and G4 RNA (NT)? (Are these the 111 and control sequences shown in Fig.3?)

- The scrambled control and PQS are Control and cMyc listed in Fig 3a, respectively. We updated the figure caption to add the missing information.

"b, c, f, Real-time intensities of transcription, translation and G4 formation assays. The constructs, non-template (NT), control (C), and template (T) are colored in blue, black, and purple, respectively. **The control and tested PQS sequences are "control" and "cMyc" listed in Fig 3a."**

2-To show that G-quadruplexes are involved in increased translation, it is necessary to test mutations that affect G-quadruplex formation. This has been done in Supplementary Fig.5. Actually, here the data sets show that G-quadruplex formation is not necessary for this phenomenon, since all mutants (M1-4) can increase translation (Supplementary Fig. 5.). While authors mention "mutated RNA can still fold into a structure that can increase translational output", these are not G-quadruplexes ! Fig.1 and supplementary Fig.5 should be merged.

Additional control sequences retaining GC content but unable to fold into G4 (this can be tested using Quadron, G4mismatch, G4hunter...) should have been tested. Despite all efforts of the authors in this work, these controls show that effects on translation do not rely on G-quadruplex formation. Authors could test if G-richness, GC content, ... are important. It is intriguing to observe that why G-rich sequences enhance translation (NT), C-rich sequences (T) inhibit it when compared to the control sequence (Fig.3).

- We thank the reviewer for providing this critical comment. First, we tested additional control sequences with higher GC content (or G-density) as shown below, which is defined by the total length of insertion. While C24 is the control used in the manuscript, C16 and C21 are new sequences tested. As shown, all three controls exhibit similar translational efficiency and we do not observe any correlation between the GC content and the translation efficiency.

Control name	Sequence	GC content	G-density
C16	GCG TGC AGG TCG AGC C	75%	44%
C21	GTC AGA CTG CTC GCT GTG TAC	57%	29%
C24	GCA TCT GTC TAG TAG TAC ACG TCG	50%	25%

- To further examine the effect of GC content or G-density more systematically, we tested additional mutant sequences (GAG and GAA series) with or without the upstream hairpin (10hp) structure (table, figure a-c below). Our new results demonstrate that all the mutants, including the [GAG 1-4] and [GAA 1-4] enhance the translational efficiency up to ~5-6 fold above the control (figure d, below). This enhancement level is equivalent to that induced by cMyc, which is the G4 with the shortest loop length (i.e. least bulkiness) and the same level of enhancement induced by two hairpins (Supplementary Fig. 3). Although these mutant sequences are not expected to form a stable G4, they are expected to fold into secondary structures that contribute to translational enhancement to a moderate level.
- By contrast, the G4s with longer loop sequences (used in the original manuscript) lead to significantly higher translational enhancement reaching up to 7-12 fold above the control as a function of loop length (figure d, right column). We plotted all the data together in the below figure for a direct comparison. This shows that within the sequence space we

examined, G4 with long loops are responsible for the extremely high, over ten-fold enhancement of translation. Despite the comparative bulkiness, consecutive hairpins, a combination of short-looped G4 and a hairpin, or random secondary structures cannot give rise to the high level of translation efficiency seen in bulky G4 structures.

- We added the following section to supplementary figure 6 in the revised manuscript. The GC content and G-density are defined by the total length of insertion sequence (16bp), resulting in the same value coincidentally.

Mutant name	Sequence	GC content	G-density
cMyc	GGG T GGG TA GGG T GGG	75%	75%
GAG 1	GAG T GGG TA GGG T GGG	68.8%	68.8%
GAG 2	GAG T GAG TA GGG T GGG	62.5%	62.5%
GAG 3	GAG T GAG TA GAG T GGG	56.3%	56.3%
GAG 4	GAG T GAG TA GAG T GAG	50%	50%
GAA 1	GAA T GGG TA GGG T GGG	62.5%	62.5%
GAA 2	GAA T GAA TA GGG T GGG	50%	50%
GAA 3	GAA T GAA TA GAA T GGG	37.5%	37.5%
GAA 4	GAA T GAA TA GAA T GAA	25%	25%

- In addition, we demonstrate here that the total loop length and the total length but not the GC-content or G-density is correlated with the translational efficiency.

- It is interesting that C-rich sequence inhibits translation efficiency. However, unlike the G-rich sequence, we don't find clear sequence-dependence in C-rich constructs. One possibility is that the C-rich sequence folds into an i-motif structure although i-motif only forms in acidic conditions. Future study is warranted.

2- While the effects of loop size on translation are clear (Fig.3), constructs that show the least potential and least stable G-quadruplexes have the strongest effects. This again object to the role of G4 in this process. In this assay, the most stable G-quadruplexes (111, cMyc and 222) show the same translational efficiency as the control sequence. The authors should conclude from this series of experiments that the formation of G-quadruplex is not involved in translation enhancement.

- As we demonstrate in the manuscript (and above), the translation efficiency is best correlated to the total loop length within the G4 forming sequences. Previous literatures define the G4 folding propensity in the DNA, but it is unclear if the same applies to RNA. Hence, we tested the folding and unwinding of both 111 and 199 RNAs, and they both show stable folding.
- We performed two orthogonal experiments to test the formation of RNA G4. First, we applied **NMR** to test if PQS in long RNA lengths of 94 and 109 nt used in our translation assay forms RNA G4. We show that they indeed form RG4 structures, evidenced by the NMR signals in 11-12ppm range due to the unique Hoogsteen base pairing of guanine imino protons. The data indicates that both cMyc (least loop length) and 199 (longest loop length) fold into RG4 (figure a, below), and the G4 signal confirms the flexibility of the structure, supporting the effect of bulkiness. Next, we applied **reverse transcriptase stop** assay. The result shows that all the RG4s induce truncated extension, indicating a similar steric effect across varying RG4 (figure b, below).
- These results are included as supplemental figure 2.

a

b

3- The authors must show in fig.2a and fig.3 the presence and absence of hairpin. Fig4.b should also show the control without hairpin.

➤ We updated all the figures as the reviewer suggested (shown below).

4- Effects on mRNA stability measured by qPCR should be represented as RNA quantity using standard qPCR quantification methods. As presented, this experiment could not be interpreted. I can see at least 1 or 2 cycles of difference between the curves (i.e. about 2 to 4 fold changes!). As a control, the amount of RNA at T0 should also be quantified by qPCR.

- We apologize that the original graph did not have the standard quantification. We clarify that since it is *in vitro* translation, there is no RNA at the T_0 . Hence, we collected the post-translational RNA after 3-hr incubation and compared the amount of RNA among control, T and, NT. Below, we updated the graph with Cq number instead of intensity curve to show there is no significant difference among the tested RNAs.
- Similarly, we also updated the supplementary 4d by quantified transcription efficiency by RT-PCR.

Figure 4b

Supplementary 4d

5- The DHX36 experiment is very elegant, but it is very difficult to distinguish between effects of DHX36 binding or G4 unwinding. The authors should have tested DHX36 without ATP.

- We updated the Supplementary Fig3b with DHX36 without ATP condition, which shows a constant low FRET state (shown below).

6- The authors tested the in vivo effects of G4-putative sequences on translation, they should also have tested series of *E. coli* 5'UTR sequences containing RG4 and controls with mutations altering the G4 structures. However, previous in vitro experiments show that G4 structures are not involved. Controls of NT sequences with mutations altering G4 formation are also missing.

- We understand the reviewer's concern about the *E. coli* 5'UTR RG4. Previous genome-wide analysis across prokaryotes by Rawal P. *et al.* (2006) and Kaplan O.I. *et al.* (2016) has revealed that DNA G4 motifs are enriched in promoter and intergenic regions. Study by Shao X. *et al.* (2020) has also demonstrated that endogenous RG4 mainly forms at coding region and can both up- or downregulate translation in *E. coli*. However, these studies report that 90% of G4 motif in prokaryotes are 2 quartet and less than 10% is G-rich (higher than 40%) G4 motif. in *E. coli*. Hence, the G4 forming sequences (3 quartet) we tested are beyond the average G4 found in *E. coli*. And we introduced the PQS further upstream of RBS than previous study (Holder IT, Hartig JS (2014)), which is another difference from the endogenous arrangement.
- Overall, our study focuses on the biophysical role of the bulky 5'UTR RG4 structures in driving the translational enhancement rather than defining the role of endogenous RG4 in *E. coli*. The constructs used in our study can be adopted by the research community to have a tunable expression of desired protein of interest. We have revised the manuscript to emphasize this point more strongly.

Other comments

a- The graphical summary of the methodology (Fig.1a) should indicate the detection methods (i.e. Beacon for transcription; GFP for translation).

- We have added the detection methods as following figure.

b- Supplementary fig.1d, Intensity and FRET tracks of a control sequence that cannot fold into G4 should also be shown.

- We have updated the Supplementary Fig1 with a new FRET histogram of polyU50 RNA to demonstrate the FRET distribution of unfolded RNA.

c- Supplementary fig.2b CMNT should be called CMYC.

- CMNT means cMyc-NT. The figure legend is updated.

d- Fig.3b needs sequence labels.

- We added the sequence labeled in Fig 3b, but we also want to indicate the curve shown here is one of representative from multiple independent repeats. The statistical quantification is addressed in Fig 3c.

e- Supplementary fig. 1b and Supplementary fig.4a-c need legends.

- We have modified the Supplementary Fig 1b and 4a with labels. However, since most of the curves in Supplementary Fig 4b and 4c are overlapped, it is not clean to label all the curves. We only use them to show the data normalization, and the quantified data is plotted in Fig 6e.

Supplementary 1b

Supplementary 4a

f- Is the difference between competitors 5 and 6 on translation significant (Fig.5d)?

- We have modified Fig 5c and 5d with quantification results and colored the competitor 5 and 6 with the same color as RBS. The statistics tests in Fig 5c and 5d are examined to control and cMyc-NT, respectively. We also updated the figure legend to make it clear.

- Yes, there is a significant difference between competitor 5 and 6 with p-value: 0.001034. The statistics tests are provided in the updated supplementary table as well as Source Data file.

g- The effects of the G4 structure could have been tested in assays with K⁺ or Li⁺ ionic conditions that stabilize and destabilize G4 formation respectively.

- We agree with the reviewer that the effect of G4 can be emphasized by comparing it to destabilizing condition (Li⁺ ionic). Unfortunately, the in vitro translation experiment was done with commercial kits, which we are unable to change the buffer conditions.

REFERENCE

1. Rawal P, *et al.* Genome-wide prediction of G4 DNA as regulatory motifs: role in Escherichia coli global regulation. *Genome Res* **16**, 644-655 (2006).
2. Kaplan OI, Berber B, Hekim N, Doluca O. G-quadruplex prediction in E. coli genome reveals a conserved putative G-quadruplex-Hairpin-Duplex switch. *Nucleic Acids Res* **44**, 9083-9095 (2016).
3. Bartas M, *et al.* The Presence and Localization of G-Quadruplex Forming Sequences in the Domain of Bacteria. *Molecules* **24**, (2019).
4. Shao X, *et al.* RNA G-Quadruplex Structures Mediate Gene Regulation in Bacteria. *mBio* **11**, (2020).

REVIEWERS' COMMENTS

Reviewer #1 (Remarks to the Author):

I appreciate that the authors have made an effort to carefully respond to the reviewers' comments. Nevertheless, I maintain that evidence of physiological relevance of the findings is essential for this study to be appropriate for Nat. Commun. I do not think that the G4-hairpin constructs are a "take-away" of broad interest. Experiments to test whether a G4 structure has a regulatory effect on an endogenous gene would not be difficult, despite the excuses that "the translation control mechanism in mammalian cells differs from E. coli" and "the mechanisms through which G4 structures can affect E. coli translation are unknown". Straightforward experiments to address this last point would make this a much stronger contribution.

Reviewer #2 (Remarks to the Author):

The authors have adequately addressed my concerns and have returned a strengthened manuscript.

Reviewer #3 (Remarks to the Author):

The authors have adequately addressed all of my concerns and comments. I have no further suggestions to improve this very nice paper.

Reviewer #4 (Remarks to the Author):

Overall, the authors provide more data, which significantly improves the manuscript. The authors did an impressive work responding to the reviewers. I feel that the manuscript will be of great interest to many readers. However, I still have one concern that I feel could be better addressed, and that lies in the involvement of the G4 structure in enhancing translation. In the new attempts presented in Supplementary Figure 6, control sequences that cannot fold into G-quadruplexes, with GAA or GAG mutations introduced in the context of the CMYC G-quadruplex, still show similar results of the G-quadruplex. Effects of the GAA, GAG and CMYC constructs are of equivalent levels and similar to two hairpins. Authors propose that these results reflect that the mutated RNA can still fold into a structure that can increase translational output. To me, this strongly confirms that it is bulkiness rather than the folding of G-quadruplex within the 5'UTR that acts in translation enhancement described here. I suggest adding a sentence or two to the discussion to address this point.

To conclude authors should be congratulated for their rigorous responses to reviewers: the manuscript is much improved

POINT BY POINT RESPONSE (2nd round)

We thank the reviewers for taking the extra time to point out remaining issues to clarify and analyze further. Please see below how we address the issues raised by the reviewers. Our answers are colored blue, headed by an arrow.

Reviewer #1 (Remarks to the Author):

I appreciate that the authors have made an effort to carefully respond to the reviewers' comments. Nevertheless, I maintain that evidence of physiological relevance of the findings is essential for this study to be appropriate for Nat. Commun. I do not think that the G4-hairpin constructs are a "take-away" of broad interest. Experiments to test whether a G4 structure has a regulatory effect on an endogenous gene would not be difficult, despite the excuses that "the translation control mechanism in mammalian cells differs from *E. coli*" and "the mechanisms through which G4 structures can affect *E. coli* translation are unknown". Straightforward experiments to address this last point would make this a much stronger contribution.

- We acknowledge the reviewers' comments on the importance of the physiological relevance of our findings. As described in our previous comment, since the location and sequence of *E. coli* G4s vary drastically from our constructs, we found it difficult to pick an endogenous gene that can accurately mimic our designed constructs. Our study aimed to perform a more biophysical analysis of how different G4 structures can influence translation enhancement. Future directions would include inserting these G4 forming sequences into an endogenous gene at the same distance away from the RBS as our *in vitro* constructs.

Reviewer #4 (Remarks to the Author):

Overall, the authors provide more data, which significantly improves the manuscript. The authors did an impressive work responding to the reviewers. I feel that the manuscript will be of great interest to many readers. However, I still have one concern that I feel could be better addressed, and that lies in the involvement of the G4 structure in enhancing translation. In the new attempts presented in Supplementary Figure 6, control sequences that cannot fold into G-quadruplexes, with GAA or GAG mutations introduced in the context of the CMYC G-quadruplex, still show similar results of the G-quadruplex. Effects of the GAA, GAG and CMYC constructs are of equivalent levels and similar to two hairpins. Authors propose that these results reflect that the mutated RNA can still fold into a structure that can increase translational output. To me, this strongly confirms that it is bulkiness rather than the folding of G-quadruplex within the 5'UTR that acts in translation enhancement described here. I suggest adding a sentence or two to the discussion to address this point. To conclude authors should be congratulated for their rigorous responses to reviewers: the manuscript is much improved

- We appreciate the reviewer has reconsidered our revised manuscript and data. Indeed, we agree that a stable structure with a certain bulky size may be able to enhance the translation.
- We have added sentences in discussion to address the point.

“...the mutation constructs still showed a similar enhancement effect as the original cMyc RG4 (Supplementary Fig. 6), reflecting that the mutated RNA can still fold into a structure that can increase translational output. Taken together with the dual-hairpin result (Supplementary Fig. 2), translational enhancement up to 4-6 folds increase occurs regardless of the structure, but the strongest effect of over ten-fold enhancement requires both stable folding and large size of the structures, for example, 177, 555 and 199 with a hairpin. **In other words, our data suggests that bulkiness of the upstream structure is more essential than the type of structure.**”

“... our study opens a wide window of opportunity for future studies, for example, **investigating the role of a stable and bulky structure, such as pseudoknot, positioned either upstream or downstream of RG4 or in between RG4 and RBS.**”